



# The Impact of CCN Concentrations on the Thermodynamic and Turbulent State of Arctic Mixed-Phase Clouds

Jan Chylik[1], Stephan Mertes[2], and Roel A. J. Neggers[1]

[1]Institute for Geophysics and Meteorology, University of Cologne, Germany.
[2]Leibniz Institute for Tropospheric Research (TROPOS), Leipzig, Germany

**Correspondence:** Jan Chylik (jchylik@uni-koeln.de)

**Abstract.**

Impacts of aerosol on mixed-phase cloud evolution play a potentially important role in Arctic climate, but remain poorly understood. The way in which aerosol, clouds and turbulence interact, is speculated to significantly modify the cloud evolution. There has been an increasing number of field observations of the ice clouds in Arctic, however it has proven hard to gain insight into these complex interactions using measurements alone. This model study aims to help filling this gap in the current understanding of low-level Arctic clouds, by combining high resolution simulations with new field campaign data. The main focus is on the impact of the cloud condensation nuclei concentration (CCN) on the properties of cloud and mixed-layer turbulence in an evolving boundary layer. We configure semi-idealised model scenarios based on the weather situation observed over open ocean during two research flights of the ACLOUD campaign, which took place over Fram Strait northwest of Svalbard. A demi-Lagrangian frame of reference is adopted, with the model domain following low level air masses and the large-scale forcings derived from weather model analyses and short-range forecasts. Adjustments in the initial state are made based on comparison to dropsonde data. The simulations reproduce the observed general structure of the cloud-bearing Arctic mixed layer. Results further show that while the ice phase forms just a fraction of the mass of cloud water, it is responsible for most of the precipitation, in line with previous observational and LES studies. A lower initial CCN concentration generally results into a faster glaciation of the cloud, leading to a faster removal of the cloud water, and also affects the vertical structure of turbulence. Implications for radiative studies of clouds for the purpose of Arctic Amplification are discussed.

## 1 Introduction

The Arctic has experienced since early 1990's more prominent warming than the rest of the world (Walsh and Crane, 1992) (Wendisch et al., 2013). This phenomenon of Arctic Amplification (Serreze and Francis, 2006) has generated a great amount of interest (Holland et al., 2003) (Overland et al., 2016) (Graversen., 2016). While low-level mixed-phase clouds are abundant in the Arctic climate, they significantly affect the surface radiative budget (Tsay et al., 1989) (Liu et al., 2017). Therefore, they are expected to play a very significant role in the Arctic Amplification (Kay et al., 2016). While the widespread melting of sea-ice and opening of leads is likely to lead to changes in the amount and the composition of mixed-phased clouds (Morrison et al., 2012) (Jun et al., 2016) (Chernokulsky et al., 2017), these changes are likely to create feedback loops due to changes in





the radiative properties of clouds (Tan and Storelvmo, 2019) (Morrison et al., 2019). Although a general increase in the cloud cover is expected (Morrison et al., 2018), the presence of larger ice particles can ultimately enhance the warming rather than reduce it (Vavrus et al., 2011) (Tan and Storelvmo, 2019).

While the importance of the mixed-phase clouds in the radiative balance is established (Morrison et al., 2011), there are
various challenges in representing these clouds in numerical weather forecast (NWP) models and climate models (Forbes and Ahlgrimm, 2014) (Kim et al., 2016) (Korolev et al., 2017). Modifications in the representation of clouds can often lead to strong changes in the magnitude of warming (Kay et al., 2016). The models are particularly sensitive to representation of ice phase and optically-thin clouds (Simjanovski et al., 2011) (Cesana et al., 2012) (Hashino et al., 2016). The currently used parameterizations of cloud processes are often based on better explored systems from lower latitudes, which are likely not very
representative of the situation in the Arctic (Vihma et al., 2014). These parameterizations then often underperform (Barton et al., 2014) (Hashino et al., 2016) (Cassano et al., 2017) (Pithan et al., 2018) (Lacour et al., 2018) because of the lack understanding of the physical processes and interactions (McIlhattan et al., 2017). Not only the parameterizations of mixed-phase clouds here do not yield suitable results, the underlying scientific knowledge of these clouds is often insufficient. This is partially caused by the lack of suitable measurements in the area, as well as biases in the satellite observations (Schweiger et al., 2002)
(Blanchard et al., 2014) (Khanal and Wang, 2001).

The need for better understanding of mixed-phase clouds and aerosols has motivated an increasing number of field observations focused on the low-level clouds in the Arctic (Curry et al., 1997) (Curry et al., 2000) (Zamora et al., 2017) (Jones et al., 2018), which have been soon followed by modelling studies (Brümmer, 1999) (Khain et al., 2001) (Gryschka and Raasch,
2005) (Klein et al., 2009) (Stevens et al., 2018). There has been also steadily growing number of studies that examine the impact of microphysical processes on the evolution of Arctic clouds (Morrison et al., 2011) (Fridlind and Ackerman., 2018) (Norgren et al., 2018). Some studies have assumed fixed values of cloud droplet number and cloud ice number concentrations (Young et al., 2018) (Kaul et al., 2015). Other have considered the explicit nucleation of cloud droplets in cloud ice (Jackson et al., 2012) (Solomon et al., 2018) (Fu et al., 2019) with rates depending on the concentration of aerosols acting as cloud
condensation nuclei (CCN) and ice nucleation particles (INP).

While the aerosols in lower latitudes modify cloud fraction, optical depth of warm clouds (Gryspeerdt et al., 2016), and also the vertical velocity (Dagan et al., 2016), including cloud-aerosol interactions generally improves the representation of warm clouds (Sotiropoulou et al., 2019). However the situation is Arctic is more complicated due to the thinner cloud layers and
co-existence of ice and liquid phase (Verlinde et al., 2007). Furthermore, there is often a complicated system of microphysical processes in the sub-cloud layer (Qiu et al., 2018). In general, aerosols in the Arctic exhibit a significant effect on cloud fraction, as well on precipitation (Zamora et al., 2018). There are indications that higher number of aerosols lead to decrease in precipitation (Lance et al., 2011), while the low aerosol concentrations limit the growth of clouds (Stevens et al., 2018).





Simulations based on Arctic field campaign data are often simplified in key aspects. This is particularly true for cloud-aerosol interaction — on one hand, the importance aerosol in microphysical processes has been strongly indicated (Zamora et al., 2017) (Ickes et al., 2018). On the other hand, model studies usually do not take into account a high variability in aerosols concentrations, which are common in the Arctic (Kalesse et al., 2016) (Willis et al., 2019). The aerosol composition and cloud properties often vary with the direction that the air originates in (Qiu et al., 2018). Aerosols are often transported to Arctic from

distant sources (Sand et al., 2017) and undergo changes due to chemical and precipitating processes (Norgren et al., 2018). The modification of aerosol composition due to cloud processes has been observed (Várnai and Marshak, 2011). Yet, these phenomena are rarely considered in the construction of model cases.

    In this paper the impact of CCN concentrations on the evolution of mixed-phase clouds and turbulence is investigated us-
ing a combination of observations and Large-Eddy Simulations (LES). Two semi-idealised Lagrangian cases are constructed based on observations during the *Arctic CLoud Observations Using airborne measurements during polar Day* (ACLOUD) field campaign, which took place in the Fram Strait in Spring 2017. One case describes a weak Cold Air Outbreak (CAO), the other a cloud situation over open water. Measurements used to construct the cases include dropsonde profiles and in-situ aerosol datasets. Unlike many previous LES studies, the number concentration of cloud droplets and the concentration of cloud
condensation nuclei are not prescribed parameters but are prognostic. This allows us to take into account the transport of CCN and their consumption by precipitating processes. With this modelling system the evolution of mixed phase clouds during the two cases is investigated, with the aim of gaining more insight into interactions between aerosol, clouds and turbulence in the Arctic. The relatively rapid growth of clouds in both case cases allows us to evaluate impacts of microphysical properties on relatively short timescales.


    Studying Arctic clouds in the Spring season is motivated by their significant sensitivity to aerosol concentrations (Fan, 2013) and the size of ice particles (Mioche et al., 2017). Also, in this season the ice phase tends to exhibit a weak dependency on the temperature, and often features high quantities of supercooled droplets (Cox et al., 2014). Moreover, significant variations in aerosol concentrations are typically observed during Spring (Willis et al., 2019). With decreasing sea-ice in a warming Arctic
climate such variations are likely to increase (Bigg and Leck, 2008), which in turn is also affected by clouds themselves (Cox et al., 2016).

    The first part of the paper focuses on the ACLOUD field campaign, providing information about the weather situation in general, the airborne and ground measurements, and the flight days selected for simulation. The construction of the semi-idealised
model scenarios is then described in detail, including the model setup and the microphysics scheme, as well as the method adopted to analyse the model results. Various sensitivity runs on model resolution and other parameters are also addressed. Next results are presented of the control runs and its evaluation against dropsonde observations, followed by an interpretation of sensitivity runs on the initial CCN concentrations. For all experiments the impact of microphysical processes on the budget of cloud water is investigated in detail, as is the behaviour of turbulence and entrainment. In the presentation of the results both





cases are always compared to assess general behaviour. The discussion focuses among others on the choice of investigating CCN over INP, and on possible limitations of this study. Finally some concluding remarks are given, and the implications of the obtained results for future studies is briefly discussed.

## 2   Observations

### 2.1   Field Campaign

The ACLOUD field campaign took place from 23 May to 26 June 2017 in the vicinity of Svalbard. Together with its sister campaign, Physical feedbacks of Arctic planetary boundary level Sea ice, Cloud and AerosoL (PASCAL), that were organised as part of the ongoing *Arctic Amplification: Climate Relevant Atmospheric and Surface Processes and Feedback Mechanisms* $((\mathcal{AC})^3)$ research program (Wendisch et al., 2017). Both campaigns provided valuable observations of Spring-time lower tropo-

sphere. ACLOUD was characterised by collocated airborne observations performed by aircraft Polar 5 and Polar 6 (Wendisch et al., 2018). In the following paragraphs, we will first briefly summarise specifics of this campaign. Secondly, we will focus of the choice of field days for further observations. Then we will discuss the observations of aerosols concentrations and their importance as a proxy for estimating the range of CCN concentrations.

The time span of the ACLOUD campaign covered three distinct regimes in synoptic conditions (Knudsen et al., 2018). The first campaign week was characterised by the advection of cold and dry air from the north. This resulted into prevailing low-level clouds over open water. During the following two weeks, the prevailing air masses originated from south and east. The lower troposphere was significantly warmer and moister, with variations in the cloud cover. The final 2 weeks were then affected by air masses from west, which lead to further variations in cloud cover and temperatures in the mid-troposphere.

Airborne observations were performed in both in stable stratified and convective conditions, in a shallow single-layer clouds, multi-layer clouds, but also in clear-sky conditions.

### 2.2   Flight Days

From the number of weather situations observed during the ACLOUD, we have carefully selected two cases of of clouds driven

by convection (see Figure 1). While each of these cases on its own is an interesting example of low-level clouds, together they provide more robust insight into the evolution of mixed-phase clouds near the sea-ice margin in Spring in the Arctic. In the following paragraphs, we will briefly describe the chosen flight days and provide justification for their selection for further investigation.



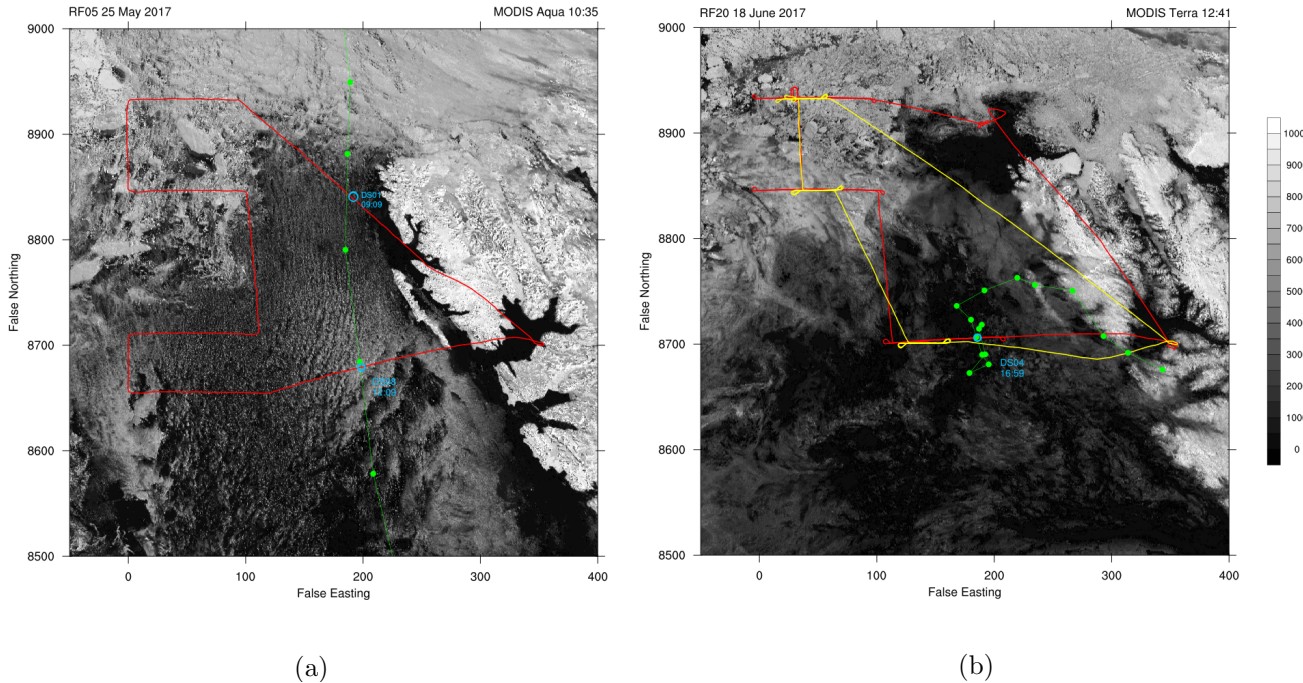

**Figure 1.** Mesoscale weather situation during a) RF05 and b) RF20 — The Svalbard land mass shows up in bright white on the right, while the sea ice is visible as grey areas in the top-left and top. The flight track of Polar 5 (red), Polar 6 (yellow), and the trajectory of the simulated domain (green) overlaid over the MODIS band 1 reflectance. The ACLOUD dropsondes (DS) used for case configuration are indicated in blue, indicating their UTC time of release. (Data was obtained through NASA Earthview)

The mission day **RF05** was concerned with the observation of a cold-air outbreak (CAO) developing over Fram Strait northwest and west of Svalbard. The advection of the cold air on 25 May 2017 then leads to a development of a typical CAO scenario (Chechin and Lüpkes, 2017) west of Svalbard. The convective rolls started forming over marginal sea-ice zone (MIZ) northwest of Svalbard. This was followed by the formation of cloud streets, which continued downwind for more than 400 km.

During RF05, the Polar 5 released dropsondes at various locations, including shallow boundary cloud streets not far of the edge of ice, as well as the deeper cloud streets further south. The backward projection of air parcel trajectories (to be discussed in section 3.3) suggests that dropsonde DS08 and dropsonde DS01 of the edge of ice approximately sampled the same air mass. This provides us with a unique opportunity to address the bias in the boundary layer depth and temperature in the ECMWF forecast. The correction of the initial state of the boundary layer is described in section 3.4.


The mission **RF20** took place on 18. June 2017 over the open water of Fram Strait west of Svalbard. The prevailing synoptic-scale situation was similar to the situation in preceding days. However there had been a slight decrease in the humidity of the





advected air in mid troposphere, resulting into disappearance of cirrus clouds from previous days (Knudsen et al., 2018). RF20 differs from RF05 in that i) the air mass is much slower moving (see Figure 1) and ii) the air mass originates over open water

already and is much warmer.

Overall, RF05 presents an illustrative example of a CAO with a gradual deepening of the Arctic mixed layer (AML). There is a single layer of low-level clouds and the atmosphere above is virtually cloud-free. This allows us to investigate a relatively simple case of mixed-phase clouds in initially cold and dry air that is then modulated by convection and radiative cooling.

RF20 then provides a "counterexample", a warmer weather situation with a weak convection is dominated by a thicker layer of low-level clouds that continue deepening.

## 2.3 Aerosol Measurements

The instruments during the ACLOUD campaign included probes with the aerosol mass spectrometer for sampling submicron-

size particles aboard the Polar 6 aircraft (Ehrlich et al., 2019). The ultra-high sensitivity aerosol spectrometer (UHSAS) measured the number size distribution of particles with diameters between 60 nm and 1000 nm by detecting scattered laser light. For further description we refer to Cai et al. (2008). The observations of aerosols particles were often collocated with the cloud radiative measurements (Wendisch et al., 2018) — i.e. on some of the flight legs over the Fram Strait, Polar 6 with aerosol probe and Polar 5 with radiative instruments flew parallel to each other at different altitudes.


Due to various technological and logistical challenges faced by Polar 6 during the ACLOUD campaign, aerosol measurements are unfortunately not available for chosen flight days. However, an extensive airborne sampling of aerosols was performed less than 44 hours after RF05, during RF07 on 27 May, when the weather situation was still dominated by the advection of the same air masses (Knudsen et al., 2018). Similarly, the large-scale weather conditions for RF20 on 18 June were very

similar to those on a previous day during RF19 flight, which also shared a similar flight path. While it is possible that the distributions of aerosols have changed, we expect a similar range of aerosols concentrations.

During RF07 and RF19, Polar 6 sampled aerosols at various altitudes. This included both horizontal flight legs at altitudes between 60 m and 3000 m, as well as profiling flight legs. However, the probe was usually not recording in the clouds due to

danger of icing. Most of the trajectory was in the free atmosphere, while the lower legs were performed only in the specific areas of interest. Aerosols sampling from 17 June (see Figure 2**b**) shows a wide range of aerosol concentrations. While the range of concentrations of aerosol particles of sizes 100–150 nm is increasing with altitude, the concentration of particles larger than 150 nm show similar ranges at all recorded altitudes.

Measurements from 27 May indicate slightly different situation. Figure 2**a** shows generally low concentrations of aerosols in the bottom 1400 m meters, followed by a wide range of concentrations above. To distinguish between the properties of





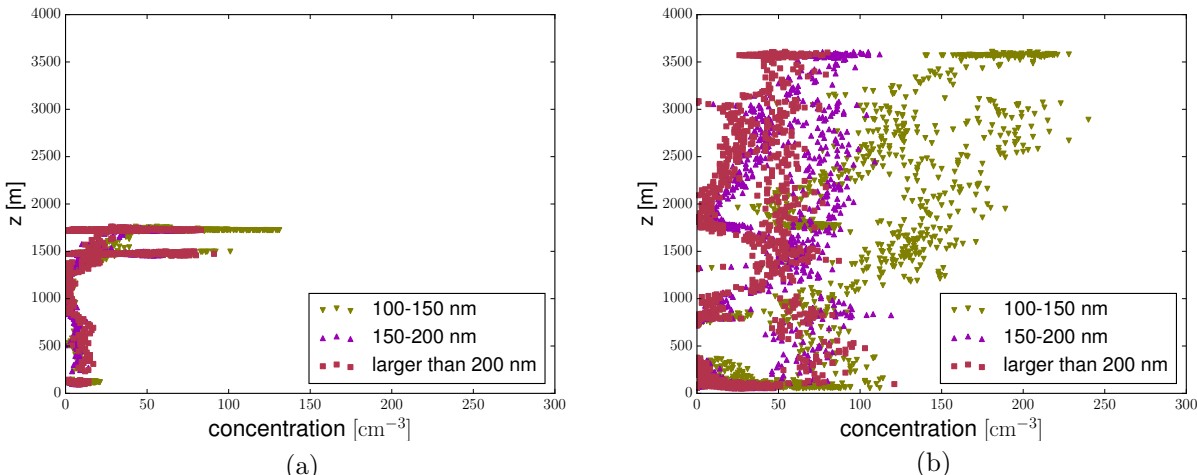

**Figure 2.** Observed aerosols concentrations from aerosol mass spectrometer aboard Polar 6 (Mertes et al., 2019) at different altitudes on a) 27 May and b) 17 June. Recorded particles were divided into size categories: 100–150 nm, 150–200 nm, and larger than 200 nm. Size categories are in the scatter plots distinguished by colour.

atmosphere in different areas, we evaluate the statistical distribution in each of them. The ranges of aerosols concentration shown in the figure 3 are then utilised in the setup of model ensembles (3.5).

## 3   Methodology

The observed weather scenarios serve as starting point for the construction of semi-idealised LESs. With the aim to follow the evolution of Arctic clouds in the cold outbreak, we follow the trajectory of an air parcel in a demi-lagrangian frame of reference (Neggers et al., 2019). The initial conditions, large-scale forcings and surface conditions are set to closely follow analyses, short-range forecasts and and observations.

The description of Methodology is divided as follows: firstly, we explain the general framework for semi-idealised LES with a demi-lagrangian frame of reference. Secondly, we focus on the bulk microphysics scheme that is has been extended for aerosols. Then we provide further information on the setting of the model ensembles.

## 3.1   DALES

The LES software package used in this study is the Dutch Atmospheric Large Eddy Simulation (DALES), formulated in Heus et al. (2010). DALES has been successfully employed in number of model studies of boundary-layer clouds in subtropics (De





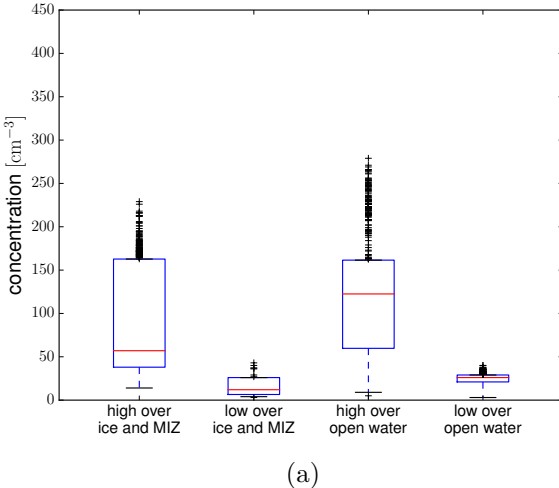
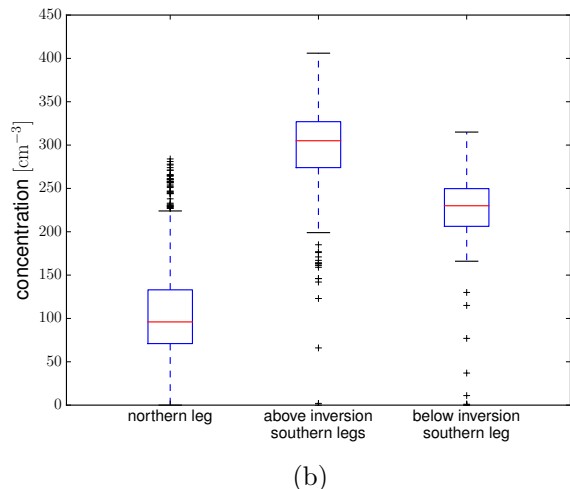

$$\text{(a)} \qquad\qquad\qquad\qquad \text{(b)}$$

**Figure 3.** The statistical distribution of observed aerosols concentrations under various conditions on a) 27 May and b) 17 June. The recorded timeseries of aerosols concentrations were divided into segments based on the area and the altitude of the flight leg they were recorded in: a) data from 27 May are divided into segments: over the sea-ice and MIZ, as well as over the open water in the middle part of Fram Strait; both at altitudes below and above 1400 m; b) data from 17 June are divided into segments: over the northern part of Fram strait, and over southern/central part of Fram strait; further divided for below the strong inversion present at altitude 1500–1900 m, and above it. Boxes indicate interquartile range of the observed distribution of aerosols, while whiskers indicate standard 1.5 range.

Roode et al., 2012), mid-latitudes (Corbetta et al., 2015) (van Laar et al., 2019), as well as polar areas (Neggers et al., 2019). It was also widely used and in studies of neutral boundary layers (Ouwersloot et al., 2016) as well as stable boundary layers
(Tomas et al., 2015). Furthermore, the scalability of DALES allow it to be utilised for simulating with various domain sizes (Griewank., 2018) DALES has taken part in various model intercomparison studies, such as stratocumulus-cumulus transition (Dussen et al., 2013) (de Roode et al., 2016) and the CONSTRAIN experiment (de Roode et al., 2019) of the GRAYZONE project.

**3.2 Demi-lagrangian Frame of Reference**

As recently argued by Pithan et al. (2016) and Neggers et al. (2019), the main motivation for doing adopting a Lagrangian configuration in the Arctic is that air masses evolve very slowly, in an area where dense networks for estimating boundary conditions and lateral forcings is totally absent Firstly, a model domain with a separate conditions on the inflow and the outflow would require detailed boundary conditions for the whole time period. Secondly, a clear benefit of Lagrangian setup is that
lateral advective forcings become zero in the air mass of interest, which significantly reduces the impact of uncertainties in the estimate of such forcing terms. Thirdly, a spatial domain that would include both the area over sea-ice as well as downwind over the open water would be extremely large for any practical computational purposes. With an aim to follow the evolution of





convective clouds in the Arctic, the model is set in a demi-lagrangian frame of references.

In this demi-Lagrangian study we follow an 'air parcel' that is advected from over a cold surface (e.g. sea ice) to over a relatively warm surface, e.g. MIZ or open water. The word "demi" here refers to the fact that the whole domain moves with the low level flow; this means that above the mixed-layer inversion, the movement vector can differ from the actual wind at those altitudes. The spatial (and temporal) changes in the surface condition along the trajectory of the moving air parcel are replaced by the temporal changes in surface conditions with time. Similarly, spatial changes in advective tendencies in the

free atmosphere along the trajectory are replaced by large-scale tendencies dependent on time and altitude. The Lagrangian approach has been widely used outside the Arctic — there have been various examples of "small domain simulations in a frame of reference moving with the mean wind have been used to explore some aspects of the evolution of roll convection" (Liu et al., 2004). Despite some degree of simplification, model studies with periodic boundary conditions and a moving frame of reference have demonstrated their usefulness in the studies of convective systems (Richardson et al., 2007).


In this study, we implement demi-Lagrangian approach similar to the method used for subtropical marine cloud transitions by Bretherton et al. (1999). An issue of this method is that it does not capture the effect of the different advection speeds due to shear within the AML. However, this issue is of a minor importance when the AML is well-mixed. Although the aforementioned approach was originally used for the boundary–layer transition in tropics, it has recently also been applied

for high-latitude conditions. Examples include, amongst other simulations of a large CAO over Gulf stream where the wind velocity is not constant with height (Skyllingstad and Edson, 2009), or simulations of shallow Arctic mixed layer (Neggers et al., 2019).

### 3.3   Back-trajectories

The starting step in the configuration of a case with moving frame of reference is determining the movement of the investigated air parcels. The dropsonde launches during RF05 and RF20 serve as a starting time and location for back-tracking the parcel movement. While the focus is on the air masses in the lower troposphere, we follow the air mass at the level 950 hPa upstream based on large-scale NWP data. The large-scale data is obtained from the products of Integrated Forecasting System (IFS) of the European Centre for Medium-range Weather forecasts (ECMWF).


For more details on this approach, we refer to Neggers et al. (2019), and will only briefly be summarised here. A combination is used of IFS analyses (available every 12 hours) and short-range forecasts (available every 3-hours), which effectively yields a four-dimensional dataset of the atmospheric state variables at 3-hourly temporal resolution and 0.1x0.1 degree spatial resolution. Horizontal advective forcing is represented through prescribed advective tendencies, calculated through horizon-

tal averaging within a $0.5° \times 0.5°$–wide column around the location of interest. The forcings are estimated at points along the trajectories, with the wind velocity at 950 hPa being subtracted from the wind profiles in the calculation of the advective



tendencies. Vertical large-scale advection is represented using a prescribed subsidence profile, by which advection becomes interactive with the simulated vertical gradients. The subsidence profile is linearized between the thermal inversion $z_i$ and the surface (Sandu and Stevens, 2011) (Neggers et al., 2017) (Loewe et al., 2017) (Sotiropoulou et al., 2018).


### 3.4 Adjustment of Initial Conditions

Most weather prediction and climate models tend to overestimate the boundary-layer depth over the sea-ice (Makshtas et al., 2007) (Jakobson et al., 2012) (Wesslén et al., 2014) (de Boer et al., 2013) (Lindsay et al., 2014). While the trajectory of air parcel in RF05 case starts over sea-ice, it likely suffers from the same problem. With a goal to correct for the possible bias in model

scenario, we focus on the independent observational data. The back trajectory of air masses from dropsonde DS08 is passing in the vicinity of previously released dropsonde DS01 (see Figure 1). This allows us to confront the thermal inversion heights extracted from the ECMWF forecast with the dropsonde sampling. The correction of the inital thermodynamic profiles and the surface conditions over the sea-ice follows the iterative method of "microgrids" described in detail in Neggers et al. (2019). Figure 4.a shows that the modelled altitude of inversion is significantly higher than in the dropsonde soundings DS01 and DS08.


For the RF05 case the initial conditions are adjusted as follows: Based on the observed inversion height at DS01 the height of the inversion in the initial state is lowered to 500 m. Below that height the temperature is reduced by 4 K, again inspired by comparison to DS01 and DS08, while the total specific humidity is adjusted such that the relative humidity is conserved within the mixed layer. Furthermore, between the old and new mixed layer height the temperature gradient is extrapolated downwards,

humidity is set constant at free tropospheric values, and any cloud condensate is removed (Neggers et al., 2019). Finally, the skin temperature over sea-ice is similarly adjusted by -4 K. After this correction, the inversion heights generally agree with said soundings (see Figure 4.b).

### 3.5 Microphysics

The DALES code was modified to include the extension for the full mixed-phase microphysics based on Seifert and Beheng (2006a). The scheme then treats subsaturation and supersaturation by the standard saturation adjustment which takes place after the collection and conversion microphysical processes (Seifert and Beheng, 2006b). This parameterisation scheme has been utilised in a number of numerical models of atmosphere. These are for example LESs such as UCLA-LES (Ackerman et al., 2009) or DALES, and mesoscale models such as ICON (Heinze et al., 2017), albeit in a modified form. A number of

implementations of this parameterization scheme has been slightly simplified. A common adjustment is such that clouds are represented by only one moment while the cloud droplet number concentration is defined as a constant parameter depending on the environment (Ackerman et al., 2009) (de Roode et al., 2019). This is also the case of the current default version of DALES, where only the warm part of the microphysics scheme is included (Heus et al., 2010).





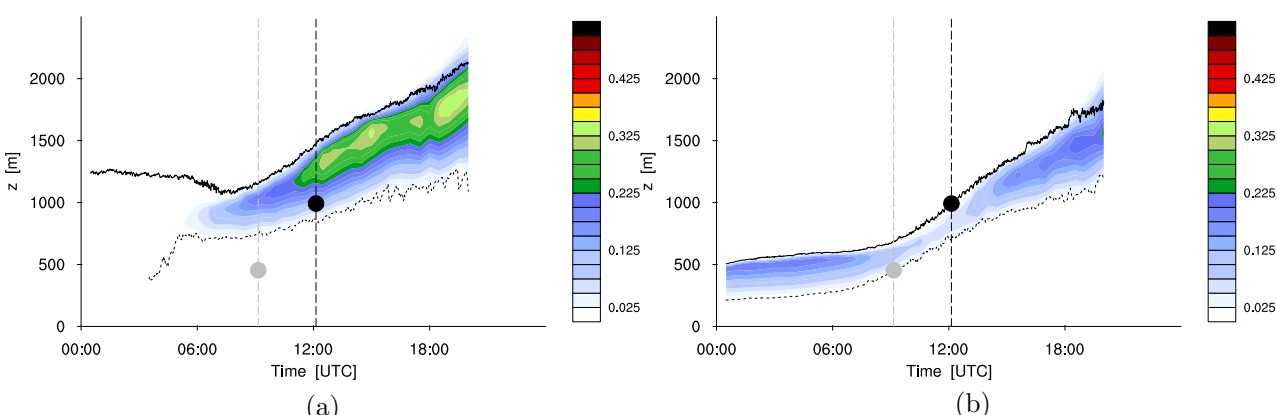

**Figure 4.** The comparison of microgrid runs, a) initialised by the original weather forecast data, and b) after the correction. The grey line indicates the launch of the dropsonde DS01, while the grey dot indicates the altitude of the steepest gradient in the potential temperature sounding. The black line and black dot then indicate the launch of DS07 and the steepest gradient there. Contourplots indicate modelled $q_{cl}$, the specific liquid cloud liquid water content, in $\mathrm{g\,kg^{-1}}$.

In this modified version of DALES, the formulation of the the 2-moment bulk scheme is closely followed in the extension – cloud droplet number is treated as a prognostic variable and the cloud nucleation is calculated explicitly. We further enhance this scheme by treating the CCN number concentration as a prognostic variable. The activation of CCN is calculated prognosti-cally in the saturated grid cells. The CCN concentration is conserved during the processes of nucleation of cloud droplet, their condensational growth and evaporation. The self-collection of cloud droplets and precipitating processes then act as a sink for

CCN. In the absence of a more advanced CCN formulation that would be compatible with 2–moment scheme, the collection of cloud droplets by ice particles and the freezing of cloud droplets are for simplicity treated as simple loss terms. Nevertheless, the glaciation of clouds here does not lead to depletion of CCN, since majority of smaller cloud droplets evaporate due to the decrease in water vapour pressure caused by the vigorous growth of ice crystals.

The value of the $\kappa$ parameter in the CCN activation relation is set to the standard value for maritime conditions (Khain et al., 2001). The microphysical parameters $a$,$b$, $\alpha$, $\beta$, $\mu$ and $\nu$ are set to the values presented in table 1. The majority of parameters follow the standard setting of Seifert and Beheng (2006a) with the exception of the values of the parameters for cloud ice, which were adjusted to the same values as in the model intercomparison study of clouds in cold climates (de Roode et al., 2019) to better suit the conditions. The diversion from the standard setting is justifiable since the original setting was based on

the high-altitude clouds in storms in temperate climates (Seifert and Beheng, 2006b).

     The choice of the the initial values of CCN concentrations is constrained by the observation aerosols 2.3. In both cases, we assume that between 50 % and 90 % observed in the parcel starting area can act as CCN. In the case of RF05, the parcel is starting in the are over ice and MIZ. The starting are of RF20 is below the inversion in the southern leg. While we do not want





**Table 1.** Overview of setting microphysical parameters for hydrometeors

|  | $a$ | $b$ | $\alpha$ | $\beta$ | $\gamma$ | $\nu$ | $\mu$ |
|---|---|---|---|---|---|---|---|
| cloud droplets | 0.124 | 1/3 | $3.75 \cdot 10^5$ | 2/3 | 1 | 1 | 1 |
| raindrops | 0.124 | 1/3 | 159.0 | 0.266 | 1/2 | $-2/3$ | 1/3 |
| cloud ice | 0.217 | 0.302 | 41.9 | 0.36 | 1/2 | 1/3 | 0 |
| snowflakes | 8.156 | 0.526 | 27.7 | 0.216 | 1/2 | 1 | 1/3 |
| graupel | 0.190 | 0.323 | 40.0 | 0.230 | 1/2 | 1 | 1/3 |

The overview covers the setting of microphysical parameters for the size as velocity of hydrometeors, as well as for particle mass distribution of hydrometeors under the assumption of generalised gamma distribution.

**Table 2.** Overview of the setting of model ensembles

|  | setting | | scenario | |
|---|---|---|---|---|
|  | $n_{CCN,\mathrm{ini}}[\mathrm{kg}^{-1}]$ | $n_{c,\mathrm{ini}}[\mathrm{kg}^{-1}]$ | RF05 | RF20 |
| ccn20 | $20 \cdot 10^6$ | $18 \cdot 10^6$ | x |  |
| ccn40 | $40 \cdot 10^6$ | $36 \cdot 10^6$ | x |  |
| ccn60 | $60 \cdot 10^6$ | $50 \cdot 10^6$ | x | x |
| ccn100 | $100 \cdot 10^6$ | $50 \cdot 10^6$ | control | control |
| ccn200 | $200 \cdot 10^6$ | $50 \cdot 10^6$ | x | x |
| ccn250 | $250 \cdot 10^6$ | $50 \cdot 10^6$ |  | x |

Each x indicate a model runs with the prescribed initial concentration of CCN and cloud droplets.

to introduce additional variability due to vertical changes in aerosol concentrations along the parcel trajectory, we set the initial CCN concentration constant per unit of mass with altitude, with the reference values at 850 hPa. The statistical distribution of aerosol concentration shown in figure 3 then motivates the setting of initial concentrations in the model ensembles prescribed in table 2. The maximum initial cloud droplet number concentration in the supersaturated areas ise set to 90% of CCN concentration in model runs with very low CCN concentrations (ccn20, ccn60), and to $50 \cdot 10^6 [\mathrm{kg}^{-1}]$ in other runs (including the

control run ccn100). The sensitivity to the setting of the initial cloud droplet number is addressed in Appendix A1.

    Due to the relatively short time-scales involved, the additional sources of CCN are neglected. This applies both to the surface flux of aerosols, as well as production of aerosols from the decay of non-nucleating aerosols. This topic is further addressed in the discussion 5.




## 3.6 Numerical Grid Configuration

The choice of the model domain is motivated by the conditions in the Arctic troposphere. The horizontal extend of the model domain is 25.6 km in both directions. This is approximately four times more than the width of the convective rolls and other convective structures commonly observe in the Arctic troposphere (Muller et al., 1999). The vertical extent of the domain is
set to 4900 m, which allows us to capture both the AML as well as a large part of free atmosphere above it. The sponge layer is applied in the top 1 km of the computational domain to prevent the reflection from the rigid top boundary.

The horizontal resolution is set 50 m in both directions. With the focus on turbulence and microphysical processes, the vertical resolution is higher in the lower part of the model domain, starting with 25 m by the ground and decreasing with the
altitude to 60 m by the domain top. The sensitivity of model results to this setup is addressed in Appendix A2; and for further details about the model setup, we refer to the statement on code and data availability.

## 4 Results

The results section is divided into two parts. Firstly, we provide a brief comparison of the modelled AML with the observations.
Secondly, we address the question of the sensitivity of model results to the model setting. Then we focus on the impact of CCN concentrations on the evolution of the vertical structure of clouds. Finally, we investigate the impact of microphysical processes of the boundary layer turbulence. To gain a deeper insight into the cloud processes, we have extended model diagnostics by recording the mass tendencies and number tendencies in cloud processes. With an aim to keep these diagnostics consistent with model diagnostics for temperature tendencies, their values are first horizontally averaged over the whole model domain
and then time averaged over the sampling time (900 s).

### 4.1 Vertical Structure and Time Evolution

The time development of the lower troposphere in **RF05** depicted in Figure 5.a shows a nearly textbook example of a CAO. The initially shallow AML undergoes rapid deepening and warming due to increase in the surface temperature after 6 hour.
Originally thin cloud later thickens due to adiabatic cooling with increasing altitude, as well as moisture transported from the surface. The comparison of vertical profiles of horizontally averaged potential temperature and humidity with the measurement from the dropsonde shows a reasonable agreement (see Figure 6.b). The height of the AML is well reproduced, as well as the strength of the inversion both in temperature and humidity. Slight negative biases exist in both the potential temperature and humidity.






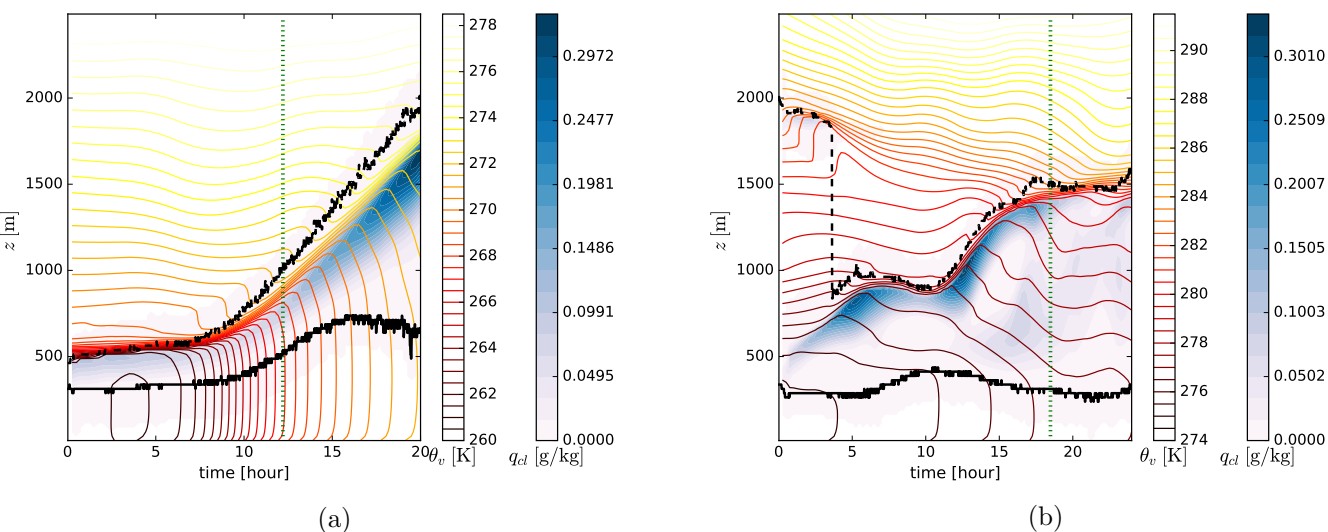

(a)             (b)

**Figure 5.** The evolution of the potential temperature and cloud liquid water content in the control runs of a) RF05 and b) RF20 scenarios. The black lines indicate cloud tops and cloud bottoms (with a threshold of $0.01\,\mathrm{g\,kg^{-1}}$ of total cloud water content), while the green line indicates the time of the dropsonde DS08.

The cloud layer significantly thickens from 10 hour onward. Over the whole time period, the liquid phase is dominating, with the ice phase contributing only 1–10 % of the total cloud water content (see Figure 8.a). However, the ice hydrometeors dominate the precipitations. At the time of dropsonde, most of the precipitation is in the form of larger ice crystals and snow. In the following hours, and increasing amount of graupel appears. This is mostly due to the thickening of the cloud (see Figure

7.a) and an increasing contribution of riming processes (see Figure 9.a). Although the precipitation in the bottom part of the clouds consists mostly of the snow, the mass of precipitating graupel and snow at the surface is approximately balanced.

The **RF20** simulation shows an evolution of thick clouds over open water (see 5.b). The main cloud deck start at the altitude 300–600 m, and continue increasing in height with time. The secondary cloud layer is initially located at the altitude 2000 m

and consist primary of ice. This secondary layer mostly evaporates and sublimates during the model spin-up, and disappears by 3 hours. For the rest of the simulation, the main cloud top is not obstructed by any additional clouds above it. With the increase in the surface temperature after 10 hour, the boundary layer enters the second warming and deepening phase. This continues until 16 hour, when the cloud top reach the altitude of a stronger temperature inversion. For the remaining 4 hours, further rising of cloud tops is inhibited, however the cloud layer continues warming due to entrainment and other processes.

Similar to RF05, the profiles of horizontally averaged potential temperature and humidity at the time of the dropsonde. Similar to RF05 the thermodynamic vertical structure shows a reasonable agreement with the dropsonde measurements (see figure 6.b).





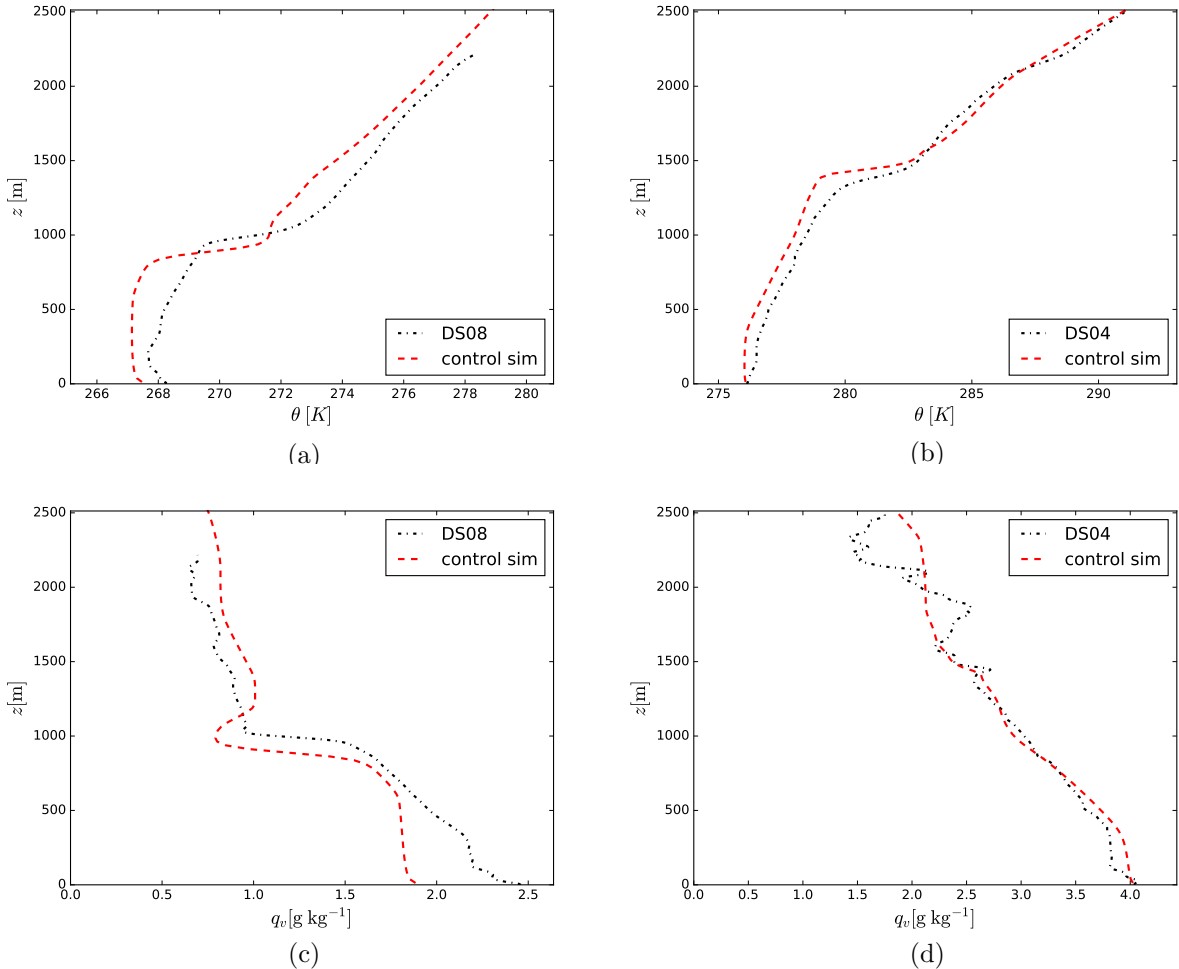

**Figure 6.** The comparison between the model with observations. The horizontally averaged profiles of potential temperature and specific humidity corresponding with the dropsonde launch are plotted against the vertical soundings from dropsondes. The comparison of the profiles from RF05 control run with the dropsonde DS08 is on the left side (a and c), while the comparison of RF20 control run and the dropsonde DS04 is on the right side (b and d).

Similarly to scenario RF05, clouds in RF20 are dominated by the liquid phase. Although liquid droplets are indicated over the whole depth of the cloud, most of the water content is located near the cloud top (see Figure 8.b. Meanwhile, the ice phase is more spread over the height of the cloud deck (see Figure 7.b). Figure 9.b shows that most of the ice precipitation forms by the aggregation of ice into snow, and then further grows by deposition. In the later stages after 15 hour, a secondary cloud deck consisting of liquid water develops in the middle part of the cloud layer (see again 5.b). At this stage, cloud ice crystals are mostly located in the upper half of the cloud layer. The aggregated snow then collects cloud droplets as it falls down.





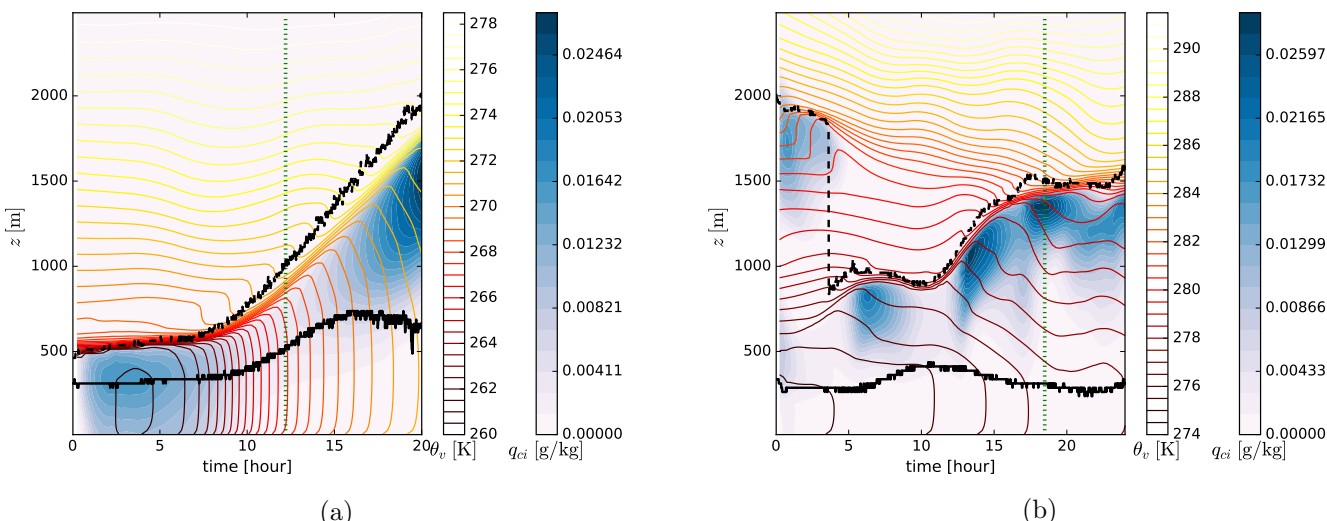

**Figure 7.** The evolution of ice cloud water content in the control runs of a) RF05 and b) RF20 scenario. The contourplots are created from horizontally averaged profiles sampled every 900 s. The black lines indicate cloud tops and cloud bottoms (with a threshold of $0.01\,\mathrm{g\,kg^{-1}}$ of total cloud water content), while the green line indicates the time of the dropsonde DS04.

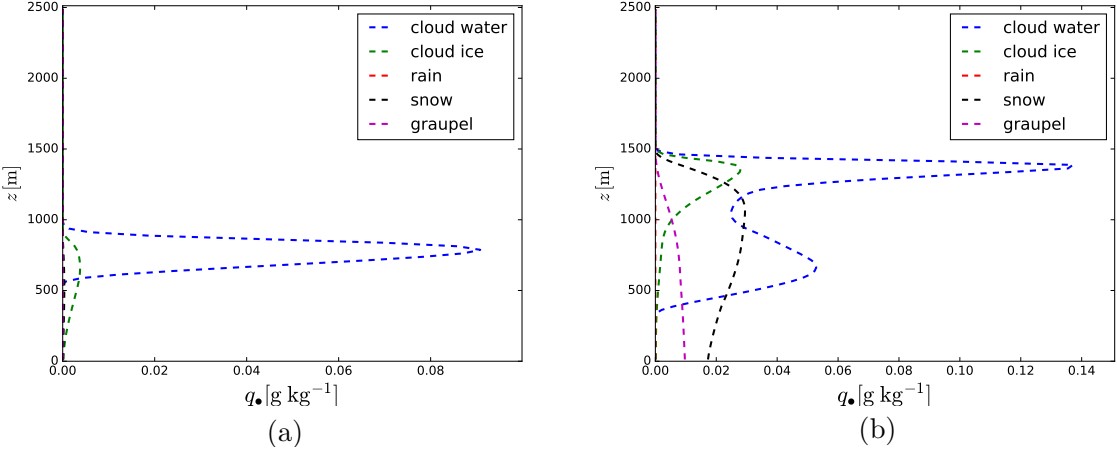

**Figure 8.** The specific mass content of hydrometeors at dropsonde launches in control runs of a) RF05 b) RF20. Lines show the horizontally averaged profiles. The value ranges on the x–axis were adapted to better show differences between the hydrometoer species, thus they differ between a and b.

## 4.2 Impact of CCN Concentrations on Precipitation

The simulations in the model ensemble of RF05 show relatively few differences during the first 12 hours. This is followed by an increasing amount of variability in the following hours. After the model spin-up, there are differences in the number of





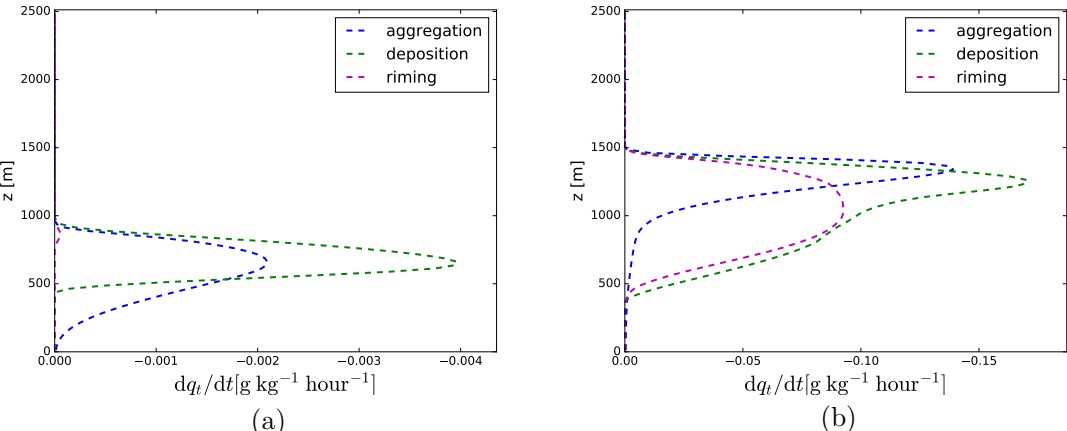

(a)                                        (b)

**Figure 9.** The specific mass content of hydrometeors and the relative contribution of processes to the formation of precipitation at dropsonde launches in control runs of a) RF05 b) RF20. Lines show tendencies that were first averaged horizontally, and then over the 900 s sampling window. The value ranges on were adapted to better show differences in the model ensembles, thus they differ between the left and the right column.

cloud droplets. These differences are however not proportional. Figure 10.a shows that in the control run and the ncc200 run, the droplet number concentration reaches 75–80 % of the CCN number concentration, only half of the CCN in ncc20 and ncc40 appears to be activated. During this stage, there are virtually no differences in the liquid path (LWP) and ice water path (IWP).

With the thickening of the cloud after 10 hours, differences in the LWP between the model runs appear and continue grow-ing (see Figure 11.a). This is primary caused by the differences in the precipitation rates (see Figure 11.e). However, there are relatively small differences in the IWP, mostly caused by the differences in the amount of snow (see Figure 11.c). The model ensemble exhibit relatively little spread in the amount of cloud ice. The aggregation of ice into snow is approximately balanced by the formation of more cloud ice. Although there are differences in LWP, all the model runs in the ensemble agree that most of the liquid droplets are in the upper part of the clouds. Furthermore, the model ensemble exhibit only slightly higher cloud tops in the runs with higher CCN concentrations (see Figure 12.a).

Meanwhile, model ensemble for RF20 scenario shows an early spread in LWP between runs (see Figure 11.b). Although the vertical structure of cloud layer remains similar as in the control run (Figure 8), there are slight differences in the height of the cloud top as well as in the amount of cloud water. Cloud tops here grow higher in the model runs with higher CCN concentrations. However the spread does not exceed 100 m (Figure 12.b). Figure 11.b further shows that differences in the LWP starts growing with the increase in LWP after 5 hour. These differences temporary disappear with the rapid growth of the clouds around 10–12 hour, but again appear soon after. While there are significant differences between the control run and runs ccn60 and ccn200, the differences between ncc200 and ncc250 mostly appear only after 18 hour. Similarly to to model





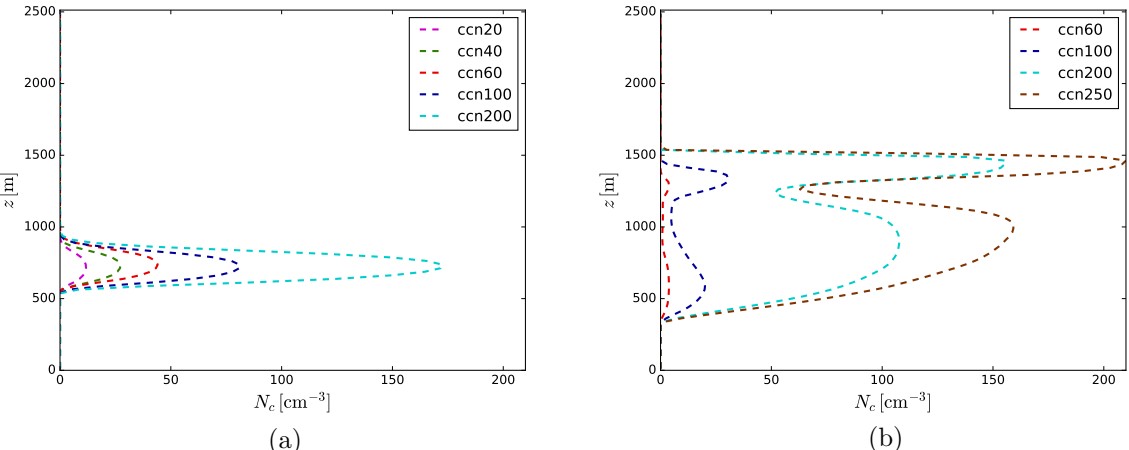

**Figure 10.** The spread in the cloud droplet number concentrations between the various model runs at the time of the dropsonde release in a) RF05 and b) RF20. Lines show horizontally averaged profiles.

ensemble RF05, runs with lower CCN concentrations generate more precipitation (see Figure 11.f).

Further evaluation of the tendencies in microphysical processes reveals significant variations between the later stages of model runs. Although ice hydrometeors still grow mostly by the deposition (see Figure 14), both model ensembles exhibit wide spread in riming of cloud droplets on ice hydrometeors (see Figure 14.b and d). In the RF05 runs, the riming tendencies are virtually negligible until 12 hours, it quickly grows afterwards. The riming rates in ncc20 and ncc40 runs are approximately twice as high as in the control run, while the riming rates in ncc200 run are by nearly order of magnitude lower. In RF05, the

riming rates in ncc60 run is slightly higher than in the ncc100 control run. Meanwhile, riming rates in ncc200 and ncc250 are usually by an order of magnitude lower. Due to favourable temperature ranges within the cloud layer the ice multiplication by Hallett-Mossop process increases the number of ice crystals in all runs of the model ensemble. However, the tendency in ice secondary ice production is in runs ncc200 and ncc250 usually by an order of magnitude lower than in the control run and ncc60.


It is important to stress that the spread in the potential temperature of AML between the runs in neither RF05 nor RF20 exceed a half of K degree. Therefore, the differences in riming rates (Figure 14) are almost exclusively caused by different sizes of cloud droplets. Changes in the size of of hydrometeor due to riming and the number of secondary ice particles than further influence other other microphysical processes. Overall, the results indicate that although the riming processes account

for less than half of the precipitation budget, they are responsible for most of the differences in LWP in the last 5 hours of models runs (see Figure 14).

**Figure 11.** The LWP, IWP, and surface precipitation in model ensembles. The left column show R05 ensemble, while the right column show RF20 ensemble. The dashed lines in c) and d) indicate the total IWP, while full lines indicates the vertically integrated water path of cloud ice crystals only.

The model ensembles RF05 and RF20 further reveal spread in the water vapour. Figure 13 shows that model runs with lower CCN concentrations generally exhibit lower water vapour content in the cloud layer. In RF20 case, the effect of CCN is

effect is strongest in the middle of the cloud layer, where is the majority of precipitation forming (see Figures 13.b and d). The



figures also show strong differences near the cloud tops, mostly due to aforementioned differences in cloud top height. The differences in the precipitation (Figure 11) affect also the rest of the boundary layer. However the effect here can be adverse for higher CCN concentrations. This phenomenon is pronounced in the last five hours of the RF05:ncc200 run (see Figure 13.c). Lower precipitation leads to less sublimation from hydrometeors, which results in lower water vapour specific content in the

bottom part of the AML. The effect of sublimation and evaporation-melting of ice precipitation in the subcloud layer slightly replenishes the water vapour there. Nevertheless, subcloud layer in RF20 case is still drier than in the model runs with higher CCN concentrations (see Figure 13.b).

## 4.3   Impacts of CCN Concentrations on Turbulence

Another important factor to consider is that there is that the impact of CCN concentrations do not only affect precipitation directly, but also involve secondary feedbacks: firstly, the decrease in CCN; secondly, the modification of turbulence due to latent heat. The first phenomena is relatively straightforward — precipitation removes water droplets as well as the activated CCN, while entrainment of the CCN-richer air from the free troposphere replenishes some of the losses. However, the second phenomenon is less straightforward, and requires further investigation.


    Figures 15.a and b show the vertical structure of TKE in both cases, illustrating that turbulence in the CAO case RF05 a lot more intense. This is due to the strong buoyancy forcing at the surface, resulting from cold air moving over warm sea water. Figure 12.d then shows that model runs with higher CCN concentrations exhibit higher entrainment velocity during the deepening of the cloud layer in RF20 model ensemble. This is also reflected in the faster the higher cloud tops in in model runs

with higher CCN (see figure 12.b). On the other hand, higher CCN concentrations in RF05 result in only minor increase in the entrainment velocities. Near cloud top, TKE is smaller with lower CCN concentrations in both cases (see Figure 15). RF05 ensemble generally show less turbulent kinetic energy (TKE) in runs with lower CCN. Figure 15 gives more information about this behaviour, showing that TKE near cloud top is smaller with lower CCN in both model ensembles. This is generally in line with the expectation that clouds with less cloud water exhibit lower intensity of processes that drive turbulence. The situation is

more complicated for the cloud layer interior, which in RF20 cases shows a reduction of TKE with higher CCN concentrations (see Figures 15.b and d). Figure 15.b shows that the TKE in ncc60 is after 5 hour generally higher than in the control run. This is followed by further increase after 15 hour. Runs with higher CCN concentration exhibit after 15 hour generally less turbulence in the cloud layer than the control runs (see figure 15.d).

The explanation of this phenomena requires further insight into the microphysical processes in the lower part of the cloud deck. The growth of the ice phase leads to the release of latent heat, warming the surrounding air. Furthermore, there are also understaturated columns where the descending ice hydrometeors begin to sublimate. The latent heat consumption during this process then further cools down the surrounding air. Higher precipitation rate then could lead to higher TKE production. The evaluation of model tendencies support this. Figure 16a shows that there is number of altitudes where a higher sublimation



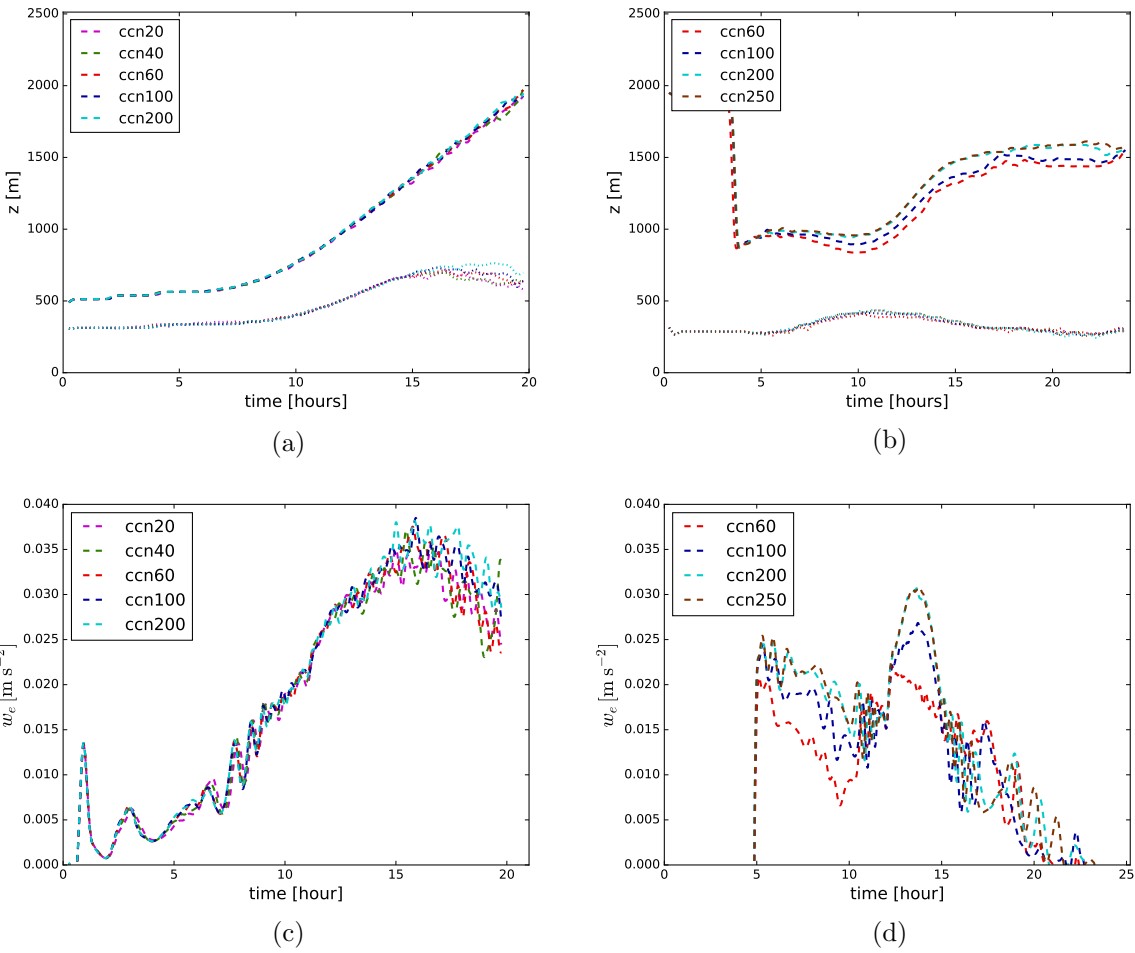

**Figure 12.** The differences in the altitude of cloud tops and cloud bottoms (a,b), and the entrainment velocity (c,d) within model ensembles. The left column depicts RF05 ensemble, while the right column depicts RF20 ensemble. The timeseries were smoothed with 1800 s moving window to remove excessive oscillations. The invalid values of model diagnostic in RF20 are not shown — in the first five hour, ther diagnostic is rendered invalid by dissipating cloud layer above, and it the last three hours, there were occasionally negative values.

rates coincide with a  positive difference in TKE. Similarly, the runs with higher CCN concentrations exhibit generally less precipitation that can sublimate and enhance the TKE (see figure 16.b,c). Nevertheless, no such effect was found in RF05 model ensemble.

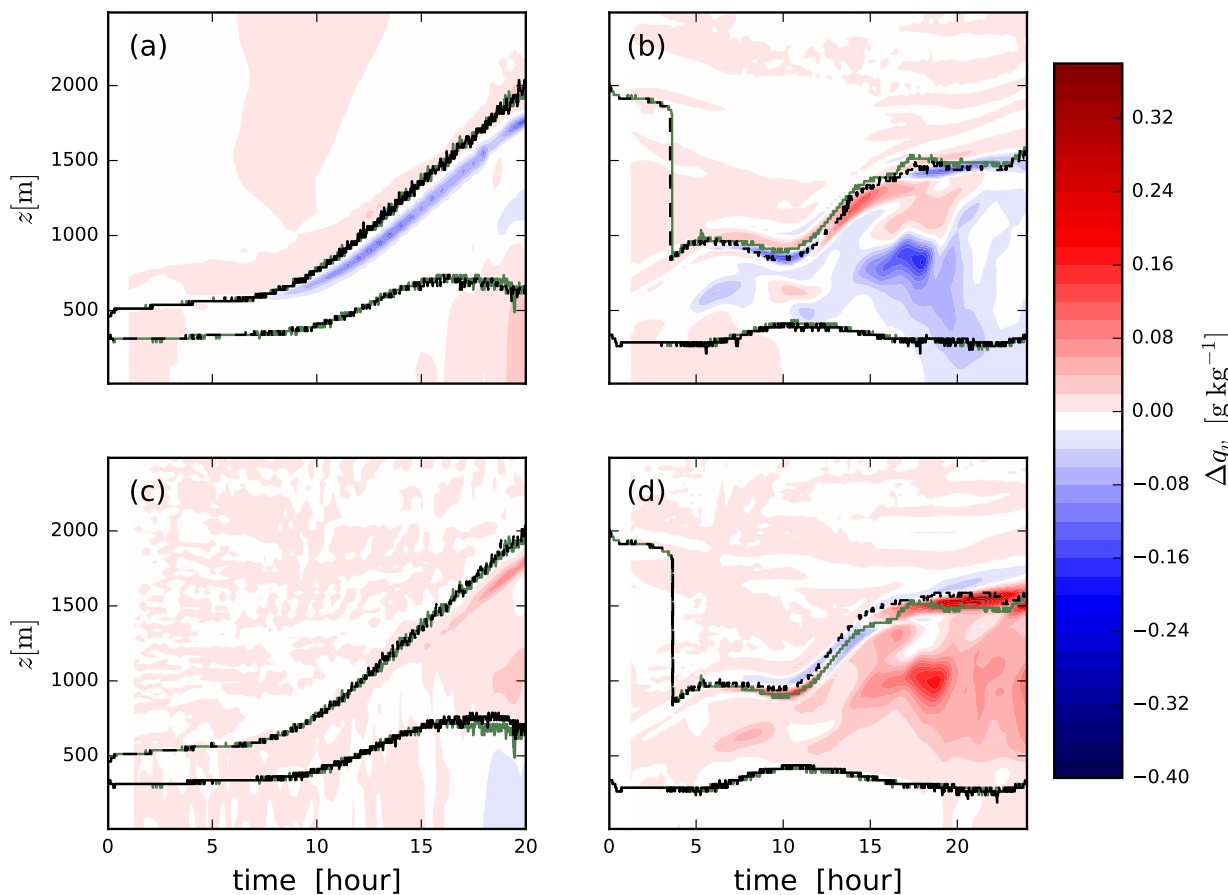

**Figure 13.** The differences in the vertical averages of water vapour between the models in the ensemble and the control runs (ncc100). The left column for RF05, while the right column for RF20. a) and b) show the TKE in control runs, c) and d) show the difference between ncc60 and control in RF05 and RF20, e) and f) show differences between ncc200 and control in RF05 and RF20. Black lines mark the altitude of cloud tops and cloud bottoms, while dark green lines mark the altitude of cloud tops and bottoms in the control run (all with a threshold of $0.01 \, \mathrm{g \, kg^{-1}}$).



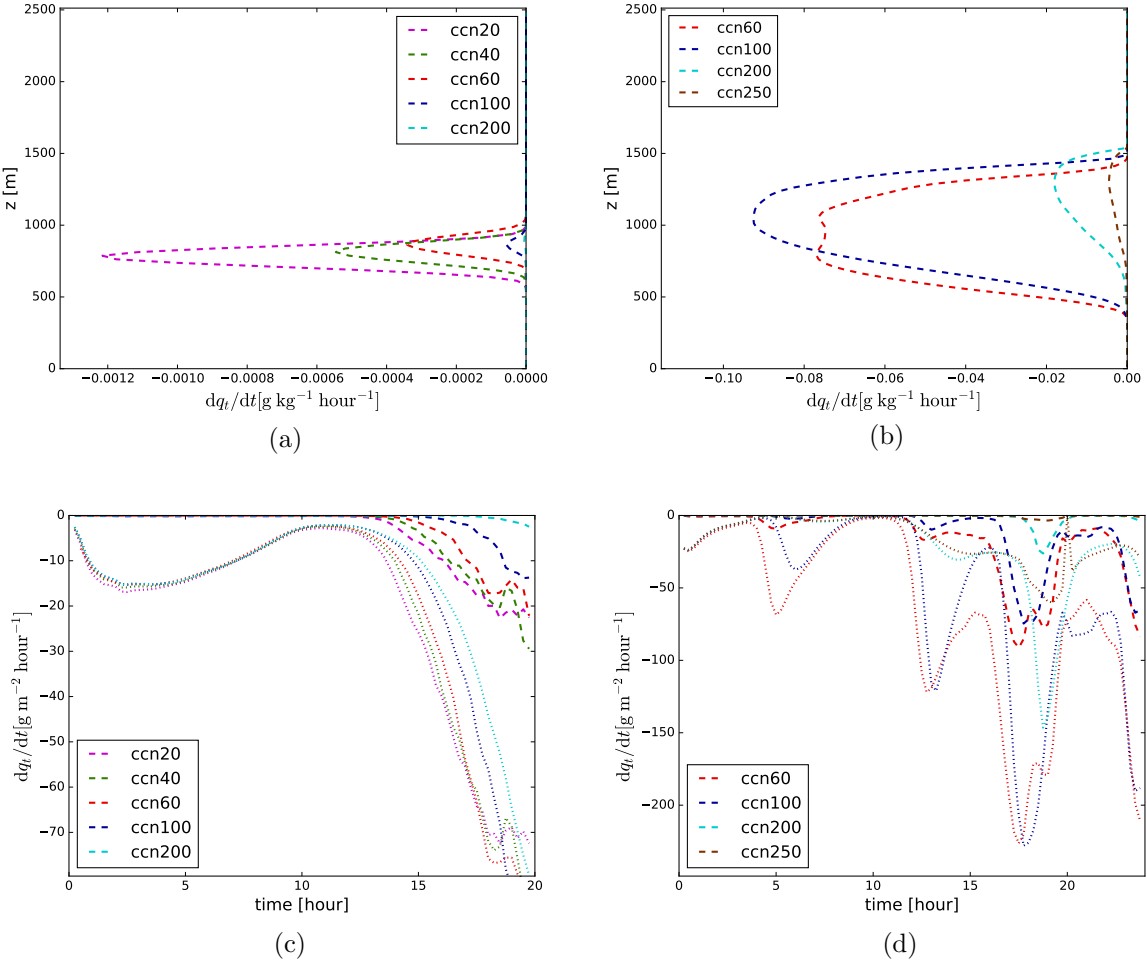

**Figure 14.** Riming tendencies in model ensembles, a)b) shows the horizontally averaged tendencies in RF05 and RF20, while c) and d) shows the time development of vertically integrated tendency. The dotted lines in c) and d) than indicate vertically integrated tendency in total precipitation production. The value ranges on the x–axis were adapted to better show differences in the model ensembles, thus they differ between the left and right column.

# 5   Discussion

## 5.1   Understanding CCN Impacts

The main goal of this study was to gain a better insight into the role of CCN concentrations in the evolution of low-level clouds and turbulence in the Arctic. LES runs follow the evolution of low-level mixed-phase clouds over open water. Both liquid and ice phase are present, with the former being dominant. Although the ice water path is only a small fraction of the total cloud water path, ice particles are mostly responsible for the removal of water from the cloud layer. In agreement with Pithan et al.





**Figure 15.** The differences in the vertical averages of TKE between the models in the ensemble and the control runs (ncc100). The left column in depicts runs from RF05, while the right column depicts RF20 runs, a) and b) show the difference between ncc60 and control in RF05 and RF20, c) and d) show differences between ncc200 and control in RF05 and RF20. Black lines mark the altitude of cloud tops and cloud bottoms, while dark green lines mark the altitude of cloud tops and bottoms in the control run (all with a threshold of $0.01\,\mathrm{g\,kg^{-1}}$).





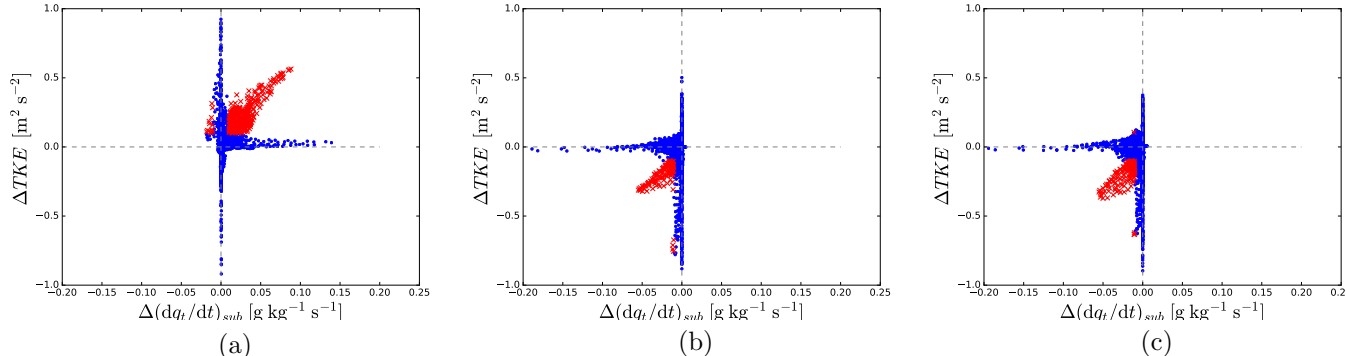

**Figure 16.** The incidence scatter plot of TKE differences with the differences in sublimation rates in RF20: a) ncc20, b) ncc200, c) ncc250. On the y–axis are differences in the horizontally averaged TKE between the run and the control run (ccn100), while on the x–axis are differences in horizontally averaged sublimation rate in the model. Each blue dot corresponds to sampling at one time at one altitude. Red cross markers then highlight points where the absolute difference in TKE exceeded threshold $0.1\,\mathrm{m^2\,s^{-2}}$ and the difference in sublimation rates exceeded $0.01\,\mathrm{g\,m^{-2}\,s^{-1}}$

(2018) and Tan and Storelvmo (2019), we find that ice cloud particles are mostly responsible for the formation of precipitation. Ice particles are mostly growing by the deposition, since saturated vapour pressure over ice surface is lower than the saturated pressure over water.

However, the mass of precipitating particles can also grow due to riming of cloud droplets. This is particularly relevant
when the cloud depth is increasing. The increased depth of the cloud increases the number of collisions between ice particles and cloud droplets, leading to higher riming rates. Deeper clouds thus allow the development of rimed snowflakes (as well as graupel), which is in agreement with the observational evidence from polar climates (Young et al., 2016). The analysis of two model ensembles indicate a clear impact of modified CCN concentrations on the precipitating processes, as well as the turbulence in the boundary layer. We are going to focus on this topic in the following paragraphs.


The CCN modifies the number of droplets, resulting into changes in ice formation. Under low CCN concentrations, precipitation tend to form easily and remove the water from the cloud layer (Mauritsen et al., 2011). Similarly to studies of Lance et al. (2011) and Zamora et al. (2017), we find that a higher CCN concentration leads to a higher number of cloud droplets, and subsequently suppressed ice formation and increased LWP. Lower CCN concentrations generally lead to faster formation
of solid precipitation, resulting into more water being removed from clouds. Still, there is also a negative feedback loop, as a removal of cloud water leads to lower amount of water available for precipitation. Overall, cloud water path differs up to the 30 % of total cloud water content. At the same time, the differences in the cloud height are relatively minor.




The explanation of these phenomena is relatively clear, but not straightforward. The composition of mixed-phased clouds is
modified by the combination of a number of microphysical processes. The rate of these process is dependent on thermodynamic
properties of the air as well as the sizes and concentration of particles involved. A larger size of a hydrometeor implies a larger
effective diameter as well as higher terminal velocity. The combination of these two increases the probability of collision with
other hydrometeors, both the hydrometeors of the same type, as well as hydrometeors of other species. Therefore, changes
in the size distribution of one hydrometeor species often lead to feedbacks in the distribution of other hydrometeor species.
For example in the case of cloud droplets, distributing the same amount of cloud liquid water over a higher number of small
droplets leads to the suppression of liquid precipitation (Seifert et al., 2015).

That said, the important phenomenon is the combined effect of mixed-phase processes. A particular case here is riming.
When cloud droplets coexist with ice particles, the increase in the size distribution of droplets results into higher collision
rates. This leads not only to the removal of said cloud droplets, but also faster increase of the mass of said ice particles. This
bears two important implications. Firstly, the ice particles grow in size and therefore descend faster. Secondly, in the favourable
temperature regime for Hallett-Mossop process (here in RF20),this also results into a stronger secondary ice production. The
higher number of newly formed small ice particles is then more likely to collide with remaining cloud droplets.

Furthermore, precipitating process can also lead to changes in the boundary layer dynamics.Ovchinnikov et al. (2011) noted
that ice deposition and sublimation can affect the buoyancy in the cloud and subcloud layer. We find that in scenarios with
deeper clouds and a relatively weak surface heating, the increased sublimation of precipitating hydrometeors coincide with an
increase in TKE. However, this effect is limited. In a case of shallower clouds and stronger surface forcing, the effect of sub-
limation is unlikely to lead to increase in TKE. We conclude that in the presence of thicker clouds, lower CCN concentration
can lead to an enhancement of boundary layer turbulence through the effect of increase sublimation of precipitation.

## 5.2  Limitations

The semi-idealised model constrained by observations presents a complex representation of developing weather situation. Nev-
ertheless, numerical simulations in general always introduce some degree of simplification of the underlying physical problem.
Simplifications are generally enforced both by the description of the physical system as well as by computational constrains.
In the following paragraphs, we will discuss the main simplifications that were introduced in our model study.

Perhaps the main simplification of our study was applying a simple parameterization of Reisner et al. (1998) for ice nu-
cleation. Although there are number of more advanced parameterizations of ice nucleation, they often require additional pa-
rameters (Fan et al., 2017). This is particularly relevant in case when we do not have a clear information about the chemical
composition of aerosol (Kanji et al., 2017) (Mei et al., 2019) or the size distribution of ice particles (Ovchinnikov et al., 2014).
Furthermore, the exact number of ice particles is generally not too important (Khain et al., 2015) for model dynamics. Then



there are also certain benefits of applying a simple scheme — it is consistent with the rest of the Seifert & Beheng parameterization (Seifert and Beheng, 2006a), and was previously used in a number of intercomparison studies (de Roode et al., 2019).


Although it would be also an interesting research question to evaluate the impact of INP concentrations in the Arctic clouds (Prenni et al., 2007), this remained outside of the scope of our study. There is a wide range in concentrations of INP (Fan et al., 2017), and a high temporal variability (Hartmann et al., 2019). But there are also a number of practical issues. Firstly, the simulations constrained with the measurements of INP generally tend to under-predict the observed amount of ice in the Arctic

troposphere (Costa et al., 2017) (Fridlind and Ackerman., 2018) and remove INPs in case of low initial concentrations (Fu and Xue, 2019) that are common in the Arctic air. Secondly, the identity of the factor that controls activation of INP is still mostly unknown (Kanji et al., 2017) (Willis et al., 2018), or is too sensitive to be easily estimated (Fan, 2013).

Furthermore, most of the ice in Arctic low levels clouds is not a pristine primary ice, but rather a mix that includes not only

frozen drops and primary ice crystals, but also broken ice pieces (Rangno and Hobbs, 2001). The origin of a wide range of particles is often related to the boundary layer circulation (Tjernström et al., 2019), although the number of particles is often higher in non-coupled clouds. The underestimation in the concentration of ice particles could be often caused by the misrepresentation of ice multiplication processes (Fridlind and Ackerman., 2018). Nevertheless, although the number of ice particles plays an important role in the AML development (Young et al., 2018), the impact of INP is not expected to be too strong (Lance

et al., 2011), and generally not stronger than impact of CCN (Gryspeerdt et al., 2018).

## 6 Summary and Conclusions

This modelling study has explored the impact of CCN concentration on the development of low-level mixed-phase clouds during the spring season in the Arctic. The main focus is on the thermodynamic and turbulent properties of developing convective

clouds. The main novelty of our study lies in the the treatment of CCN concentration as a prognostic variable and accounting for the consumption of CCN in some of the cloud processes. Semi-idealised LES scenarios are based on observed cold-air outbreak cases during the ACLOUD field campaign, and constrained with the date from airborne observations.

Our main findings are:


– While the ice phase forms just a fraction of the mass of cloud water, it is responsible for most of the precipitation. Increasing the CCN concentrations leads to the suppression of ice formation, and thus decreased precipitation. This is in line with other LES and observational studies.





– While lower CCN concentrations lead to increased precipitation, the main process responsible for the faster removal of water is riming. Although the ice particles grow mostly by deposition, riming of cloud droplets accounts for most of the differences between the the runs with different CCN concentrations.

– Although the lower CCN concentrations lead to lower liquid water path, it causes just very minor differences in the vertically integrated mass of simple cloud ice. The explanation of two-fold: firstly, increased aggregation of ice crystal, and secondly, increased collection of ice crystal by rimed snow particles.

– Differences in CCN concentrations can indirectly affect boundary layer turbulence. Turbulence (TKE) near cloud top increases with CCN in both cases, boosting top-entrainment. However, there canbe additional effects in the cloud layer interior. In the cases of a deeper cloud layer with a sufficient amount of water, lower CCN results into an increased turbulence in the boundary layer. This is due to increased amount of sublimation of ice hydrometeors that affects the buoyancy.

Overall, this study indicates an importance of the effect of CCN concentrations on the structure of low-level mixed-phase Arctic clouds and turbulence. The evolution of the cloudy Arctic mixed-layer can not be fully understood without the knowledge of the aerosols properties. Considering the high variability in Arctic air masses, the impact of CCN concentrations on the liquid water path and snow precipitation should be taken into account in parameterizing the radiative properties of said clouds for the purpose of NWP forecast and climate models. This is becoming increasingly important with the changes in the sea-ice cover in the warming Arctic. . Regarding the future observational studies in the Arctic, we encourage simultaneous measurements of microphysical and turbulent properties of the air. It is expected that this can help us to gain a deeper insight into the enhancement of turbulence by heat release due to precipitation. We are looking forward the upcoming Multidisciplinary drifting Observatory for the Study of Arctic Climate (MOSAiC) field campaign, which will include extensive amount of collocated measurements of the properties of Arctic air.

*Code and data availability.* The current version of DALES (dales-master 4.1) is available on: https://github.com/dalesteam/dales/releases/tag/v4.1, The aerosol dataset is part of the data publication on PANGEA: https://doi.org/10.1594/PANGAEA.900403, The configuration files for model scenarios, our code update, as well as the main model outputs are available at https://doi.org/10.5281/zenodo.3271773 (last update July 2019).

**Appendix A: Sensitivity Study**

There is a number of model parameters that can theoretically alter the results. Firstly, we have investigated the sensitivity to $n_{c,\mathrm{ini}}$, the initial cloud droplet number concentration. The sensitivity runs were performed for both scenarios. Secondly, the sensitivity of the model runs to the setting of grid parameters was assessed by means of an additional set of model runs for the scenario RF05. In the following paragraphs, we provide a brief description of the sensitivity tests. For further results of the





sensitivity tests, we refer to appendix.


## A1 Sensitivity to Initial Cloud Droplet Number

The sensitivity to the initial cloud droplet number concentration was investigated using a set of model runs where the CCN concentration was the same as in the control run. During the cloud initialisation, the cloud droplet number in the saturated areas was set to the maximum values a) $18 \cdot 10^6 \ \mathrm{kg}^{-3}$, b) $80 \cdot 10^6 \ \mathrm{kg}^{-3}$. The test revealed that although the a lower cloud droplet

number concentration leads to faster formation of precipitation in the first few model steps, this effect is short-lived. The differences between the model runs have disappeared already within the model spin-up in the first hour of model run. Therefore, in the initial cloud droplet number concentration is not a relevant parameter in this model study.

The test revealed that the impact of the initial cloud droplet number concentration is very short-lived. Although the a lower
cloud droplet number concentration leads to faster formation of precipitation in the first few model steps, this effect is very short-lived. In the convective boundary layer, the differences between the model runs have disappeared already within the model spin-up in the first hour of model run. There are no significant differences in the first 18 hours of model runs (see figure A1). The spread between the model runs that appear later are related to random differences in the runs. Overall, in the initial cloud droplet number concentration is not considered a relevant parameter in this model study.

## A2 Sensitivity to Domain size and Resolution

The sensitivity to the domain setting was tested on a set of model runs where one of the properties was modified. While the boundary layer in R05 scenario is shallower than in RF20 and clouds are generally thinner, it was expected that this scenario would possibly exhibit more sensitivity to the grid resolution. The sensitivity set included both the model runs with changes in vertical or horizontal resolution, as well as the model domain with changes in the horizontal extend. The main results are
summarised in the table A1.

Overall, the analysis of the sensitivity to model resolution indicated differences in TKE and precipitation. The results suggest that coarser resolution leads to oscillations in the entrainment velocity, as well as higher precipitation. The coarser horizontal resolution generally lead to decrease in TKE and lower water path. The run with final horizontal resolution produced more snow
within clouds, however there were no significant changes in the LWP nor IWP. Generally speaking, further improvements in the domain size or vertical resolutions lead to increase in the computational expenses without a clear benefit on model results.

*Author contributions.* SM provided the aerosols field measurements, as well as the guidelines on the treatment of aerosols. RN designed the model framework and prepared the model forcing files. JC developed the model extension for the interaction of aerosols and mixed-





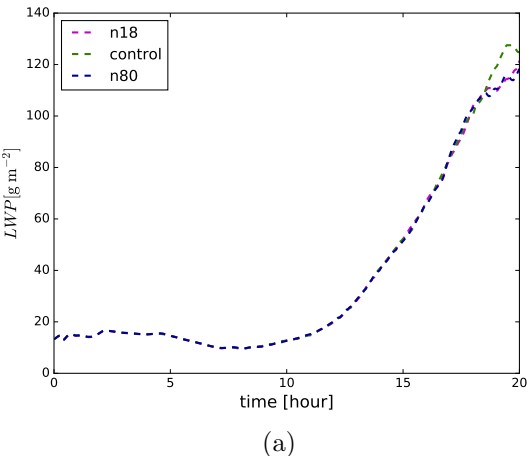

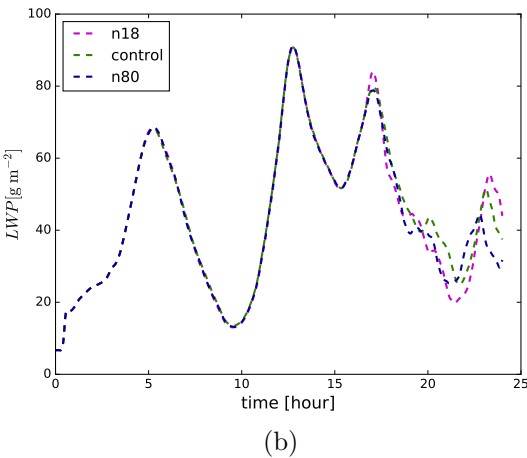

**Figure A1.** The LWP in the set for the sensitivity test to initial cloud droplet number in a) R05 and b) RF20 cases. Runs start with the same initial CCN concentration (ccn100), but different initial cloud droplet number concentrations (see A2). The value ranges were adapted to better show differences in the model ensembles, thus they differ between the left and the right column.

**Table A1.** the overview of results of sensitivity test to domain size and resolution

| modified property: | impact on: | | | |
| --- | --- | --- | --- | --- |
| | LWP | precipitation | $w_e$ | $maximum\,of\,TKE$ |
| control | $21.7\,g\,m^{-2}$ | $0.07\,g\,m^{-2}\,s^{-1}$ | $0.03\,m\,s^{-1}$ | $2.2\,m^{-2}\,s^{-2}$ |
| domain size decreased to $\frac{1}{2}$ in both directions | -8 % | +2 % | -10 % | -3 % |
| domain size doubled in both directions | +1 % | +3 % | +1 % | -1 % |
| coarser horizontal resolution: 100 m | -20 % | -15 % | +2 % | +11 % |
| finer horizontal resolution: 25 m | +9 % | +37 % | -11 % | -9 % |
| coarser vertical resolution: regular 40 m grid | +18 % | +60 % | oscillations | -11 % |
| finer vertical resolution: regular 25 m grid | -6 % | -1 % | +7 % | +5 % |

The comparison of sensitivity runs with the control run. Differences are described relative to the control run at the time of the dropsonde launch. Dashes indicates no significant differences.

phase microphysics. JC performed the model simulations and the respective analysis of model output. JC prepared the manuscript, while RN

supervised the process and revised the manuscript.

*Competing interests.* The authors declare that they have no conflict of interest.





*Acknowledgements.* We gratefully acknowledge the funding by the Deutsche Forschungsgemeinschaft (DFG, German Research Foundation) – Projektnummer 268020496 – TRR 172, within the Transregional Collaborative Research Center "ArctiC Amplification: Climate Relevant Atmospheric and SurfaCe Processes, and Feedback Mechanisms $(\mathcal{AC})^3$". The Gauss Centre for Supercomputing e.V. (www.gauss-centre.eu)

is acknowledged for providing computing time on the GCS Supercomputer JUWELS at the Jülich Supercomputing Centre (JSC). We further thank the Alfred Wegener Institute (AWI), the PS106/1 crew and the ACLOUD science teams for making the field campaign happen.





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
