# Peer review of "The Impact of CCN Concentrations on the Thermodynamic and Turbulent State of Arctic Mixed-Phase Clouds"

_Atmospheric Chemistry and Physics, 2019_

## Referee Comment (RC1)

Comments on "The impact of CCN concentrations on the thermodynamic and turbulent state of Arctic mixed-phase clouds"

This manuscript is trying to explore the impact of CCN concentration on the development of low-level mixed-phase clouds during the spring season in the Arctic. Data from the recent ACLOUD field campaign are valuable and are used to initialize the setup for the demi-Lagrangian Large Eddy Simulations. Sensitivities modeling studies are performed for two scenarios to study the effect of CCN concentration on mixed-phase clouds, precipitation, and turbulence. Aerosol-cloud interaction in the mixed-phase clouds is very important but is still far from clear. Therefore, this modeling work along with the unique observational dataset might help to bridge the gap in this field. However, the current version is not well written. I find a bunch of typos and grammatical errors in the manuscript. In addition, I find two "main" conclusions in this paper controversial: (1) A lower CCN concentration results to a faster glaciation of the cloud due to stronger precipitation (which contradicts the general idea of cloud invigoration); and (2) The increased amount of ice sublimation will increase TKE in the boundary layer (which contradicts the idea of PBL stabilization due to sublimation). Please find my major comments below, especially 1,3,7. I'm not saying that what most people, maybe just me, believe is correct. Results are interesting here, but I'm not fully convinced by the explanation. I am also confused by some results. I think the main reason is that lots of details of the model setup are not clearly described. For example, ice microphysical scheme, which is very important to the mixed-phase clouds, is not discussed at all. Without those details, it is difficult for me to understand the results and their interpolations. Therefore, I suggest at least a major revision for this manuscript.

Major comments:

1. One of the main conclusions, as stated in the abstract is that "A lower initial CCN concentration generally results into a faster glaciation of the cloud, leading to faster removal of the cloud water, and also affects the vertical structure of turbulence." I think this conclusion might lead to misunderstanding. Based on theory, e.g., Korolev and Isaac (2006), the glaciation process is very sensitive to the ice number concentration in the mixed-phase clouds, which has been confirmed in many previous LES studies (Fridlind et al, 2002; Ovchinnikov et al., 2014; Fu et al., 2019… already cited in the manuscript). However, the link between CCN concentration and ice nucleating particle (INP) concentration is still unclear. "A lower initial CCN concentration" corresponds to a lower INP concentration or same INP concentration? I think results are very sensitive to how the model takes care of the ice formation process for a lower CCN concentration, which is not clear in the abstract. For an extreme condition, if INP concentration decreases significantly faster than CCN concentration, it might lead to a condition that only a few ice particles exist in the mixed-phase clouds, and therefore, the glaciation will be slower than before.

2. Line 79: "on relatively short timescales". How short the timescale is? Within an hour? A few hours? One day? A few days? Please state it clearly.

3. Section 3.5: It is not clear to me how the microphysical scheme deals with ice nucleation? I see later that in section 5.2 (line 493): "Perhaps the main simplification of our study was applying a simple parameterization of Reisner et al. (1998) for ice nucleation." I think it is too late to make such a statement. In addition, I think it is not good enough to just cite Reisner et al. (1998). Ice nucleation is crucial to ice number concentration in the mixed-phase clouds. Ice number concentration is one of the most important variables that affect the glaciation process (see my comment 1). Please clearly state that how ice particles are formed in the model: from supercooled cloud droplets? from ice nucleating particles? Please clearly state that what controls ice number concentration in the model: fixed number concentration? diagnostic value? prognostic value? Any other process related to ice microphysics: rimming? collision? secondary ice production?... The goal is to make it clear in the manuscript that (1) how ice particles are formed in the model? (2) What controls the ice size and number concentration? (3) and most importantly, when changing CCN concentration in the model, will it also change ice nucleation and ice number concentration?

4. Figure 6: If I understand correctly, DS08 is used to adjust the initial condition for the model setup for RF05, and DS04 is used to adjust the initial condition for the model setup for RF20. If so, it is not surprising to me that those vertical profiles in Figure 6 are close to the sounding. It will be more interesting for me to see the comparison between model and observations at a later time, e.g., DS01 for RF05.

5. Another question is related to large-scale forcing (discussed in section 3). Please clearly state how large-scale forcing is taken care of in the model: the forcing profile is from ECMWF forecast? Which variable is forced: wind speed? temperature? water vapor? What is the forcing time scale? Are the forcing profiles also "corrected", similar to the adjusted initial conditions? If yes, please discuss more details that how the forcing profiles are corrected in section 3. If no, will the bias in boundary layer depth and temperature affect the results?

6. Line 380-390: The rimming rate is higher for lower CCN concentrations, which explain why precipitation is stronger and LWP is lower for relatively clean case (as shown in Figure 11). This result is consistent with the warm cloud microphysics: cleaner clouds prefer larger cloud droplets, and thus benefit rimming process and precipitation. However, my question is that will it also work for the mixed-phase cloud? There are lots of study talking about cloud invigoration by aerosols in the cold cloud, and results from those studies are in general opposite to the conclusion in this study. So the question is how general results in this study are? I guess it strongly depends on the cloud microphysical scheme. But the cloud microphysical scheme is not clearly discussed in the previous section (see comment 3).

7. Line 430-434. I'm not fully convinced by the statement here, because sublimation of ice particles, in general, occurs in the subsaturated region with respect to ice (below the mixed-phase cloud base). Sublimation in the sub-cloud layer will cool the lower part of the atmospheric boundary layer, and thus stabilize the boundary layer (smaller TKE). If what you propose is true, I don't quite understand (1) why sublimation occurs in the cloud layer and (2) why sublimation will enhance TKE physically, beyond the statistical correlations shown in Figure 16. Note that cloud top and cloud base heights are similar in the ensemble for RF05, but there is a difference in

the ensemble for RF20 (Figure 15). So is the weaker turbulence for a relatively polluted case for RF20 due to the thinner cloud layer? Another thought is that how about the effect of the temperature anomaly? The significant difference in water vapor for RF20 (Figure 13) is probably due to the temperature difference because water should be saturated in the cloud region and the saturated water vapor only depends on temperature. I think it would be nice to show the temperature as well, similar to qv (Figure 13) and TKE (Figure 15).

8. I strongly suggest the author check gramma carefully before submission. I find several typos and grammatical errors. Some of them including some minor comments are listed below.

Minor comments:

1. Line 4: change "ice clouds" to "mixed-phase clouds"?

2. When citing multiple papers at one place, please combine them in one bracket. For example, the first sentence should be "…(Walsh and Crane, 1992; Wendisch et al., 2013)" instead of (Walsh and Crane, 1992)(Wendisch et al., 2013).

3. Line 23: "is likely to", change "is" to "are".

4. Line 52-54: Rephrase the first sentence "While the aerosols…". It is not clear.

5. Line 76: change "mixed phase clouds" to "mixed-phase clouds"

6. Line 102: change "were" to "was"

7. Line 104: change "provided" to "provide"

8. Line 115: "in both" is not clear to me.

9. Line 198: missing "." after "absent"

10. Line 281: Define those microphysical parameters, at least in the Appendix.

11. Line 293: change "ise" to "is"

12. Table 2: Since for control simulation, RF05 and RF20 are the same. Why there are no ccn20 and ccn40 for RF20, and no ccn250 for RF05?

13. Line 564: delete "the" before "a lower cloud droplet"

14. Line 314: Change "two" to "three"?

15. Figure 5 caption: green line in Figure 5b indicates DS04? Similarly, in Figure 7, the green line in the left indicates DS08? It is not clear in the caption.

16. Line 352: Add "Figure" before "5.b"

17. Line 358-360: "Figure 10.a shows that…". The statement seems not correct. Figure 10 a. shows that almost all CCN are activated for ncc20 and ncc40. Please check.

18. Figure 13 caption: doesn't make sense. There is no (e) and (f). Please check.

19. Line 377: Delete one "to"

20. Figure 14 caption: "The dotted lines in (c) and (d) than…" Delete "than"

21. Figure 14 caption: Add "and" between a) and b)

22. Line 399-400: Delete "is effect".

23. Line 411: "…firstly, the decrease in CCN…" is not clear to me. "decrease in CCN" is not "the impact of CCN concentration". It is the "impact of precipitation". Please rephrase this sentence.

24. Line 419: "This is also reflected in the faster the higher cloud tops in in model runs with higher CCN" is not clear to me. Please rephrase it.

25. Line 458-459: "we find that a higher CCN concentration leads to a higher number of cloud droplets, and subsequently suppressed ice formation and increased LWP." Please discuss whether it is a general statement or relies on the model setup.

26. Figure 16: (a)(b)(c) eat some space of x label.

27. Line 536: add space between "can" and "be".

---

## Referee Comment (RC2) · Anonymous Referee #2 · 4 Sep 2019

The topic of the presented study is highly interesting and the title is quite intriguing. However, when reading and trying to understand the study and the results, I am not impressed. The manuscript is in quite a bad state with lots of repetition, spelling errors, poor quality figures, missing figure etc. This is a misuse of the time of a reviewer. In the current state, it is not possible to review the science and I therefore recommend rejection of this manuscript. I hope the authors take the time to do a thorough rewrite, I am happy to review it again provided it is presented with the details and explanations that is needed to assess the quality and validity of the work performed. Below I illustrate my decision with some examples, they are far from exhaustive though.

The title contains Arctic Mixed-Phase Clouds which in my view leads to the long-lived persistent clouds that are found in the Arctic. The simulation, however, are based on observations that are classified as cold-air outbreak days. Although the clouds in cold-air outbreaks also can be of mixed-phase, which is common in Cu clouds, the title is still a bit misleading.

The presentation of the cases that the study is based on does not give enough information to be convincingly chosen. The background information is scattered and not coherent. Figure 1 claims to present the "mesoscale weather situation" and shows MODIS Aqua views and the flight tracks. It is not even clear what clouds we are looking at and how this relates to the design of the experiments.

Figure 2, where are the observations taken?

Figure 3, there is a mixture of statistical methods in this figure, using medians and interquartile ranges are used if data is non-gaussian. Why then plot standard deviations, if that is what is meant by "standard 1.5 range"?

The explanation and motivation for the demi-Lagrangian method cannot be understood.

A statement like "has been widely used outside the Arctic" must be followed by references.

On Page 10, you write that you are referring to DS01 and DS08 but in the Figure it says DS01 and DS07.

The alterations that are done for RF05 are huge and still the comparison with the DS01 is way off although you write "generally agree". Where is the comparison of the vertical profiles? What stratification, winds, and RH do you have? Are we even close to reality? Do we have any idea if the turbulence in the LES is generated by the correct processes to be able to compare with reality and thus analyse any sensitivity?

Table 1 consists of coefficients that you do not explain at all what they are used for. Are they all unitless?

Figure 5 and others, do you really think that it is appropriate to provide four significant numbers for the cloud liquid water content?

Figure 13, caption explains panels e and f. The figure only contain a-d.

---

## Author Comment (AC1) · 24 Dec 2019

**Response to Anonymous Reviewer 2**

We would like to thank the anonymous reviewer for the careful assessment of our submitted manuscript. We appreciate the willingness to review our manuscript, and the detailed comments provided, which draw our attention the ambiguities in the description of our observations and methodology, as well as to other shortcomings of our manuscript. In the revision of our manuscript, we have tried to address each point. Please find our detailed response to each of your points in the text below.

**1 Major Comments**

The topic of the presented study is highly interesting and the title is quite intriguing. However, when reading and trying to understand the study and the results, I am not impressed. The manuscript is in quite a bad state with lots of repetition, spelling errors, poor quality figures, missing figure etc. This is a misuse of the time of a reviewer. In the current state, it is not possible to review the science and I therefore recommend rejection of this manuscript. I hope the authors take the time to do a thorough rewrite, I am happy to review it again provided it is presented with the details and explanations that is needed to assess the quality and validity of the work performed. Below I illustrate my decision with some examples, they are far from exhaustive though.

We thoroughly appreciate the assessment by the reviewer that the topic of our study is highly interesting. It is clear that the assessment of the first submitted version was quite critical. What we take away from this review is that the overall quality of the text is simply too far below the acceptable standard for scientific publication, and that this is the main reason for the reviewer to recommend a rejection. While not intended as an excuse, it is worth noting at this point that this paper is the first by the corresponding author as a lead-author, which might explain some of these deficiencies. In the revision of the manuscript we have taken great care to achieve an acceptable standard. Many changes have been made to the text, the figures, and the content. One of the main goals was to remove the inconsistencies as identified by the reviewer. We sincerely hope that these changes will be sufficient; in that sense we are grateful for the willingness of the reviewer to again review the revised manuscript.

**1.1 Title**

The title contains Arctic Mixed-Phase Clouds which in my view leads to the long-lived persistent clouds that are found in the Arctic. The simulation, however, are based on observations that are classified as cold-air outbreak days. Although the clouds in cold- air outbreaks also can be of mixed-phase, which is common in Cu clouds, the title is still a bit misleading.

The clouds that were the centrepiece of this research are really mixed-phase clouds, as is clearly evident from the measurements performed during the ACLOUD campaign (Wendisch et al., 2018). There were a number of previous studies that talk about convective mixed-phase in the Arctic, such as Fridlind et al. (2007) and Verlinde et al. (2007). While RF05 is a clear case

of a cold outbreak, case RF20 focus on a very different scenario — it is a case of multilayered cloud in the absence of rapid surface heating.

**1.2   Observed Weather Scenarios**

The presentation of the cases that the study is based on does not give enough in- formation to be convincingly chosen. The background information is scattered and not coherent. Figure 1 claims to present the "mesoscale weather situation" and shows MODIS Aqua views and the flight tracks. It is not even clear what clouds we are looking at and how this relates to the design of the experiments.

The description of the cases has been thoroughly rewritten. Firstly, we have added a motivation for the choice of cases to the introduction (lines 72-76). Secondly, with an aim to improve the clarity of the case description, we have modified the caption of the figure showing the MODIS satellite products (Figure 1). Much more information is now provided, such that these cases are better motivated and should also be reproducible by independent researchers. For example, initial profiles are now shown for multiple variables. The MODIS images are now better described and the markers for location have also been improved. In these satellite images, clouds can indeed be hard to distinguish from sea ice; however, the main purpose of including these images is to illustrate the cases the best as we can; we do not have access to better data products.

**1.3   Aerosols Observations**

Figure 2, where are the observations taken?

We have tried to improve the clarity of the description of the locations of aerosol observations. In the revised version of the manuscript, we have explicitly indicated the geographical locations of aerosol measurements (Figure 1).

**1.4   Aerosols Statistics**

Figure 3, there is a mixture of statistical methods in this figure, using medians and in- terquartile ranges are used if data is non-gaussian. Why then plot standard deviations, if that is what is meant by "standard 1.5 range"?

We thank the reviewer for pointing out that the formulation "standard 1.5 range" is ambiguous. After considering various options, we have modified the Figure 3 so the whiskers now indicate the 5th and 95th percentiles.

**1.5   Demi-lagrangian**

The explanation and motivation for the demi-Lagrangian method cannot be understood. A statement like "has been widely used outside the Arctic" must be followed by refer- ences.

Indeed, the demi-Lagrangian method is not new concept, and it has been widely used outside of Arctic. We have included many references to such studies, including Liu et al., (2004) and Richardson et al. (2007). The demi-Lagrangian configuration means that only part of the column is aligned with the flow: in our case, the lower atmosphere, where the mixed-phase clouds reside.

The benefit of this frame of reference is that it minimises the amplitude of the large-scale advective tendencies that have to be prescribed; as a result, uncertainty in the case setup is reduced. This motivation and explanation is now added to the text (lines 196–216). For further explanation of the approach, we refer to Neggers et al. (2019).

**1.6 Dropsonde**

On Page 10, you write that you are referring to DS01 and DS08 but in the Figure it says DS01 and DS07.

The typo in the legend of the figure has been corrected, it is now referring to dropsondes DS01 and DS08.

**1.7 Adjustment of Initial Conditions**

The alterations that are done for RF05 are huge and still the comparison with the DS01 is way off although you write "generally agree". Where is the comparison of the vertical profiles? What stratification, winds, and RH do you have? Are we even close to reality? Do we have any idea if the turbulence in the LES is generated by the correct processes to be able to compare with reality and thus analyse any sensitivity?

We agree that the alterations are substantial. However, this is unavoidable and a necessary evil whenever large-scale model data are used to drive and LES experiment, given the well-known significant biases in the state of the lower atmosphere The absence of dense observational networks simply leaves us no other choice. While the original Figure 4 was misleading, we have instead included overview of the initial profiles.

To meet the reviewer's demand for a comparison of vertical profiles between LES and observations, Fig. 6 has been added. Indeed the match is not perfect; but we do not agree it is "way off" either (which in itself is a subjective and this non-scientific statement). For example, i) the inversion heights are pretty close, ii) so is the free tropospheric state, and iii) the vertical structure below the inversion is similar. Only for water vapour do somewhat larger deviations exist. However, in our experience in comparing LES to observations, deviations of this magnitude are pretty typical. As previous LES studies of a similar nature have shown, a less than perfect agreement with observations does not complicate the use these simulations for gaining insight into how nature might work; as long as they are in the right ballpark. This is exactly the purpose of this study; not to achieve a perfect match with observations, but to learn from behaviour of mixed-phase microphysics in a resolved, turbulent environment that is more or less representative of the situation probed by the aircraft during ACLOUD.

Finally, concerning the comment on the generation of turbulence, in these situations of mixed-phase clouds over open water there are various sources: i) wind shear (mechanical turbulence), ii) surface buoyancy flux and iii) long-wave cooling at cloud top. All three processes are represented in the simulations; accordingly, one can conclude that the turbulence is generated by the right processes.

**1.8   Coefficients for Ice Microphysics**

Table 1 consists of coefficients that you do not explain at all what they are used for. Are they all unitless?

All coefficients in the table are now explained in the text and have the right units. We have checked the literature and added the appropriate units to the shape and size parameters. Since the part of the methodology on microphysics was thoroughly rewritten, the table is now located the appendix A1

**1.9   Figures 5**

Figure 5 and others, do you really think that it is appropriate to provide four significant numbers for the cloud liquid water content?

The organisation of the figures in the manuscript has been thoroughly modified, better grouping them around a specific topic and also leaving out figures and panels that were not essential. We have also improved the labelling of the colourbars of the contourplots. We have replaces previously arbitrary values that are multiply of 0.04 and increased the number of labelled values. In the similar manner, we have also adjusted figures for concentration of cloud ice (now Figure 5.b-c).

**1.10   Figure 13**

Figure 13, caption explains panels e and f. The figure only contain a-d.

We have corrected the text of the caption (now Figure 12) to refer to panels a-d instead of c-f

---

## Author Comment (AC2) · 24 Dec 2019

**Response to anonymous Reviewer 1**

We would like to thank the anonymous reviewer for the careful and detailed assessment of our submitted manuscript. We appreciate the constructive comments, which have been a great help in improving the manuscript in both presentation and content. In the revision of our manuscript, we have adopted vast majority of suggested modifications. Please find our detailed response to the comments below. For the purpose of clarity, we have structured our response to section for each of the major points.

**1 Major Comments**

This manuscript is trying to explore the impact of CCN concentration on the development of low-level mixed-phase clouds during the spring season in the Arctic. Data from the recent ACLOUD field campaign are valuable and are used to initialize the setup for the demi- Lagrangian Large Eddy Simulations. Sensitivities modeling studies are performed for two scenarios to study the effect of CCN concentration on mixed-phase clouds, precipitation, and turbulence. Aerosol-cloud interaction in the mixed-phase clouds is very important but is still far from clear. Therefore, this modeling work along with the unique observational dataset might help to bridge the gap in this field. However, the current version is not well written. I find a bunch of typos and grammatical errors in the manuscript. In addition, I find two "main" conclusions in this paper controversial: (1) A lower CCN concentration results to a faster glaciation of the cloud due to stronger precipitation (which contradicts the general idea of cloud invigoration); and (2) The increased amount of ice sublimation will increase TKE in the boundary layer (which contradicts the idea of PBL stabilization due to sublimation). Please find my major comments below, especially 1,3,7. I'm not saying that what most people, maybe just me, believe is correct. Results are interesting here, but I'm not fully convinced by the explanation. I am also confused by some results. I think the main reason is that lots of details of the model setup are not clearly described. For example, ice microphysical scheme, which is very important to the mixed-phase clouds, is not discussed at all. Without those details, it is difficult for me to understand the results and their interpolations. Therefore, I suggest at least a major revision for this manuscript.

In the revision of this manuscript we put much effort into improving the text, which we agree was lacking in quality in the first submission. This paper is the first as a lead author, which might explain some of these textual errors, for which we apologize.

We also spent a lot of time on elaborating on the two main conclusions as identified by the reviewer above. We agree that these insights might appear controversial, and that this justifies a careful analysis. We hope that in the new version these conclusions are presented more clearly and also better supported by the results presented.

Finally, as requested by the reviewer, the description of the setup of the model and the cases has been substantially rewritten, and now includes a detailed description of the ice microphysics scheme. Our detailed responses to the individual comments below indicate which modifications have been made.

**1.1 INP concentrations**

1. One of the main conclusions, as stated in the abstract is that "A lower initial CCN concentration generally results into a faster glaciation of the cloud, leading to faster removal of the cloud water, and also affects the vertical structure of turbulence." I think this conclusion might lead to misunderstanding. Based on theory, e.g., Korolev and Isaac (2006), the glaciation process is very sensitive to the ice number concentration in the mixed-phase clouds, which has been confirmed in many previous LES studies (Fridlind et al, 2002; Ovchinnikov et al., 2014; Fu et al., 2019... already cited in the manuscript). However, the link between CCN concentration and ice nucleating particle (INP) concentration is still unclear. "A lower initial CCN concentration" corresponds to a lower INP concentration or same INP concentration? I think results are very sensitive to how the model takes care of the ice formation process for a lower CCN concentration, which is not clear in the abstract. For an extreme condition, if INP concentration decreases significantly faster than CCN concentration, it might lead to a condition that only a few ice particles exist in the mixed-phase clouds, and therefore, the glaciation will be slower than before.

We agree with the reviewer that it is important how the model takes care of the ice formation process at lower CCN concentrations. We admit this was not as clearly described in the original version as it could have been, something which we have tried to improve in the revision. While the main focus of this study was on the effect of changes in CCN concentration, the INP concentrations were prescribed following the original setting of Seifert and Beheng (2006a). In the revised manuscript, we clarify that in the description of ice nucleation (lines 271–278) and freezing of cloud droplets. In this study, we did not include a dependence between INP and CCN concentrations.

**1.2 Timescales**

2. Line 79: "on relatively short timescales". How short the timescale is? Within an hour? A few hours? One day? A few days? Please state it clearly.

We thank the reviewer for pointing out the ambiguity of this statement. The evolution of the convective cloud field observed during ACLOUD campaign typically took place on timescales of hours to tens of hours (lines 81–82). The statement in the manuscript has been adjusted accordingly.

**1.3 Ice production**

3. Section 3.5: It is not clear to me how the microphysical scheme deals with ice nucleation? I see later that in section 5.2 (line 493): "Perhaps the main simplification of our study was applying a simple parameterization of Reisner et al. (1998) for ice nucleation." I think it is too late to make such a statement. In addition, I think it is not good enough to just cite Reisner et al. (1998). Ice nucleation is crucial to ice number concentration in the mixed-phase clouds. Ice number concentration is one of the most important variables that affect the glaciation process (see my comment 1). Please clearly state that how ice particles are formed in the model: from supercooled cloud droplets? from ice nucleating particles? Please clearly state that what controls ice number concentration in the model: fixed number concentration? diagnostic value? prognostic value? Any other process related to ice microphysics: rimming? collision? secondary ice production?... The goal is to make it clear in the manuscript that (1) how ice particles are formed in the model? (2) What controls the ice size and number concentration? (3) and most importantly, when changing CCN concentration in the model, will it also change ice nucleation and ice number concentration?

In the revised version a detailed description is added how ice particles are formed in the model (Section 3.4). We have dded a brief recapitulation of the primary and the secondary production (lines 270-288).Our enhanced implementation of Seifert & Beheng microphysics add number of CCN as a prognostic variable, but otherwise closely follows Seifert and Beheng (2006a) and Seifert and Beheng (2006b). In addition, it is now clearly stated how size and number concentrations are represented: we closely follow the original Seifert and Beheng (2006) paper. Finally, it is now explicitly mentioned that there is no direct relation between CCN and INP concentrations (lines 277-278).

**1.4 Adjusting the profiles**

4. Figure 6: If I understand correctly, DS08 is used to adjust the initial condition for the model setup for RF05, and DS04 is used to adjust the initial condition for the model setup for RF20. If so, it is not surprising to me that those vertical profiles in Figure 6 are close to the sounding. It will be more interesting for me to see the comparison between model and observations at a later time, e.g., DS01 for RF05.

The correction is performed only in the scenario RF05, where the air mass start over sea ice (lines 222–225). In the RF20 scenario, the initial point of model domain is over open water, and therefore does not require adjustment of the initial conditions. There is good agreement between the modelled and observed values, as demonstrated by figure 6.b and 6.d. In addition, our opinion it is far from trivial that the modeled and observed profiles should be close together: there is a significant time in between the initialization and the time-point at which the model is compared to the dropsondes, so there is enough time for the model state to drift. The dependence of the vertical thermodynamic structure of Arctic mixed layers on upstream conditions was explored in a recent paper by Neggers et al (JAMES, 2019). We now refer to this study to elaborate on this behavior. We hope these changes are sufficient to address the reviewer's concern.

**1.5 Large Scale Forcing**

5. Another question is related to large-scale forcing (discussed in section 3). Please clearly state how large-scale forcing is taken care of in the model: the forcing profile is from ECMWF forecast? Which variable is forced: wind speed? temperature? water vapor? What is the forcing time scale? Are the forcing profiles also "corrected", similar to the adjusted initial conditions? If yes, please discuss more details that how the forcing profiles are corrected in section 3. If no, will the bias in boundary layer depth and temperature affect the results?

We thank the reviewer for this constructive feedback, we agree the description could be more precise and complete.
In the revision of the manuscript we have put a lot of effort in improving this section (3.2). For example, references to previous studies have been added that contain detailed and extensive descriptions of the method adopted here to derive the large-scale advective tendencies from ECMWF data. Also, an equation is added that defines exactly how this tendency is adjusted for the speed of movement along the trajectory. Finally, the questions of the reviewer are answered in the text; i) what is the time-resolution of the prescribed forcing (3 hours), ii) which variables are forced, and iii) that the profiles of the prescribed advective tendencies are not adjusted.

Concerning the decision not to adjust the forcings, we argue that, because the Lagrangian transformation reduces these tendencies to a minimum at low levels anyway, no adjustment is needed. For example, the main adjustment is in RF05 case, and involves a *lowering* of the initial inversion; but below the ECMWF inversion the forcing profiles are more or less constant with

height, and very small anyway because of the Lagrangian framework. Also note that in that height range the subsidence profile is linearized anyway. As a result, the prescribed advective forcing of the mixed layer does not change much when maintaining the original forcing profiles.

We hope that these modifications and additions are effective in clarifying this section.

**1.6  Cloud Invigoration**

6. Line 380-390: The rimming rate is higher for lower CCN concentrations, which explain why precipitation is stronger and LWP is lower for relatively clean case (as shown in Figure 11). This result is consistent with the warm cloud microphysics: cleaner clouds prefer larger cloud droplets, and thus benefit rimming process and precipitation. However, my question is that will it also work for the mixed-phase cloud? There are lots of study talking about cloud invigoration by aerosols in the cold cloud, and results from those studies are in general opposite to the conclusion in this study. So the question is how general results in this study are? I guess it strongly depends on the cloud microphysical scheme. But the cloud microphysical scheme is not clearly discussed in the previous section (see comment 3).

As stated in our response to comment 1.3, we have put a lot of effort in describing the microphysics scheme in more detail. We further agree that cloud invigoration is an interesting topic. However, it is not the main topic of this study, which is how mixed-phase cloud microphysics interacts with turbulence. Perhaps invigoration plays a role in this interaction; but for now we consider this a topic for future research.

While it is true that lots of studies talk about the effect of cloud invigoration, these studies mostly deal with different kinds of mixed-phase clouds — various clouds with "warm base" and "cold cloud top" (Fan et al., 2012). Nevertheless, the review of cloud invigoration by Altaraz et al. (2014) points out that there are conflicting results for clouds with a cold base, i.e. where the cloud base is located at the freezing level or above. In the revised manuscript we briefly discuss this issue on lines 442-455.

**1.7  Sublimation and TKE**

7. Line 430-434. I'm not fully convinced by the statement here, because sublimation of ice particles, in general, occurs in the subsaturated region with respect to ice (below the mixed-phase cloud base). Sublimation in the sub-cloud layer will cool the lower part of the atmospheric boundary layer, and thus stabilize the boundary layer (smaller TKE). If what you propose is true, I don't quite understand (1) why sublimation occurs in the cloud layer and (2) why sublimation will enhance TKE physically, beyond the statistical correlations shown in Figure 16. Note that cloud top and cloud base heights are similar in the ensemble for RF05, but there is a difference in the ensemble for RF20 (Figure 15). So is the weaker turbulence for a relatively polluted case for RF20 due to the thinner cloud layer? Another thought is that how about the effect of the temperature anomaly? The significant difference in water vapor for RF20 (Figure 13) is probably due to the temperature difference because water should be saturated in the cloud region and the saturated water vapor only depends on temperature. I think it would be nice to show the temperature as well, similar to qv (Figure 13) and TKE (Figure 15).

We would like to thank the reviewer for drawing our attention to possible confusion related to this point. Our response to the two numbered questions above: (1) the sublimation occurs in the lower part of cloud later, which in the scenario RF20 includes number of undersatured (and thus cloud-free) downdraughts — in that sense, the structure of clouds here resembles the

situation of stratocumulus over cumulus. This is distinctly different from the scenario RF05. (2) The sublimation of ice hydrometeors in the undersatured patches/downdraughts cools down the air in the undersaturated areas and thus futher decreases their (already negative) buoyancy. Meanwhile, the hydrometeors falling through saturated areas do not lose mass, but rather gain mass due to deposition and riming of cloud droplets. The resulting differences in buoyancy could also enhance the amplitude of vertical motions, and with it TKE. This phenomenon has actually been identified in a number of studies of mixed-phase clouds with cold cloud base (Lynn et al., 2005; Lee and Feingold, 2010). Furthermore, hydrometeors falling through undersaturated areas can trigger turbulence even in the areas that were previously dynamically stable (Kantha et al. 2019).

With a goal to improve the clarity of the text, we have added few explanatory sentences to the discussion (lines 472-478), inserted the references mentioned above concerning the impact of deposition on negative buoyancy in downdrafts, and emphasised the limitation of this point in Conclusions (lines 520, 530 and 531).

**1.8 Typos**

8. I strongly suggest the author check gramma carefully before submission. I find several typos and grammatical errors. Some of them including some minor comments are listed below.

We have corrected all the typos and gramatical errors from the following list.

**2 Minor Comments**

Minor comments:

All minor comments have been processed into the revised manuscript

1. Line 4: change "ice clouds" to "mixed-phase clouds"?

2. When citing multiple papers at one place, please combine them in one bracket. For example, the first sentence should be "...(Walsh and Crane, 1992; Wendisch et al., 2013)" instead of (Walsh and Crane, 1992)(Wendisch et al., 2013).

3. Line 23: "is likely to", change "is" to "are".

4. Line 52-54: Rephrase the first sentence "While the aerosols...". It is not clear.

5. Line 76: change "mixed phase clouds" to "mixed-phase clouds"

6. Line 102: change "were" to "was"

7. Line 104: change "provided" to "provide"

8. Line 115: "in both" is not clear to me.

9. Line 198: missing "." after "absent"

10. Line 281: Define those microphysical parameters, at least in the Appendix.

11. Line 293: change "ise" to "is"

12. Table 2: Since for control simulation, RF05 and RF20 are the same. Why there are no ccn20 and ccn40 for RF20, and no ccn250 for RF05?

The choice of model runs with adjusted CCN concentrations is motived by the aerosol observations (Figure 3) in the free atmosphere

13. Line 564: delete "the" before "a lower cloud droplet"

14. Line 314: Change "two" to "three"?

15. Figure 5 caption: green line in Figure 5b indicates DS04? Similarly, in Figure 7, the green line in the left indicates DS08? It is not clear in the caption.

16. Line 352: Add "Figure" before "5.b"

17. Line 358-360: "Figure 10.a shows that...". The statement seems not correct. Figure 10 a. shows that almost all CCN are activated for ncc20 and ncc40. Please check.18. Figure 13 caption: doesn't make sense. There is no (e) and (f). Please check.

18. Line 377: Delete one "to"

19. Figure 14 caption: "The dotted lines in (c) and (d) than..." Delete "than"

20. Figure 14 caption: Add "and" between a) and b)

21. Line 399-400: Delete "is effect".

22. Line 411: "...firstly, the decrease in CCN..." is not clear to me. "decrease in CCN" is not "the impact of CCN concentration". It is the "impact of precipitation". Please rephrase this sentence.

23. Line 419: "This is also reflected in the faster the higher cloud tops in in model runs with higher CCN" is not clear to me. Please rephrase it.

24. Line 458-459: "we find that a higher CCN concentration leads to a higher number of cloud droplets, and subsequently suppressed ice formation and increased LWP." Please discuss whether it is a general statement or relies on the model setup.

25. Figure 16: (a)(b)(c) eat some space of x label.

26. Line 536: add space between "can" and "be".